# Convergence of $\log(1/\epsilon)$ for Gradient-Based Algorithms in Zero-Sum Games without the Condition Number: A Smoothed Analysis

**Ioannis Anagnostides**
Carnegie Mellon University
`ianagnos@cs.cmu.edu`

**Tuomas Sandholm**
Carnegie Mellon University
Strategic Machine, Inc.
Strategy Robot, Inc.
Optimized Markets, Inc.
`sandholm@cs.cmu.edu`

## Abstract

Gradient-based algorithms have shown great promise in solving large (two-player) zero-sum games. However, their success has been mostly confined to the low-precision regime since the number of iterations grows polynomially in $1/\epsilon$, where $\epsilon > 0$ is the duality gap. While it has been well-documented that linear convergence—an iteration complexity scaling as $\log(1/\epsilon)$—can be attained even with gradient-based algorithms, that comes at the cost of introducing a dependency on certain condition number-like quantities which can be exponentially large in the description of the game.

To address this shortcoming, we examine the iteration complexity of several gradient-based algorithms in the celebrated framework of *smoothed analysis*, and we show that they have *polynomial smoothed complexity*, in that their number of iterations grows as a polynomial in the dimensions of the game, $\log(1/\epsilon)$, and $1/\sigma$, where $\sigma$ measures the magnitude of the smoothing perturbation. Our result applies to optimistic gradient and extra-gradient descent/ascent, as well as a certain iterative variant of Nesterov's smoothing technique. From a technical standpoint, the proof proceeds by characterizing and performing a smoothed analysis of a certain *error bound*, the key ingredient driving linear convergence in zero-sum games. En route, our characterization also makes a natural connection between the convergence rate of such algorithms and perturbation-stability properties of the equilibrium, which is of interest beyond the model of smoothed complexity.

## 1 Introduction

We consider the fundamental problem of computing an *equilibrium* strategy for a (two-player) zero-sum game

$$\min_{\boldsymbol{x} \in \Delta^n} \max_{\boldsymbol{y} \in \Delta^m} \langle \boldsymbol{x}, \mathbf{A}\boldsymbol{y} \rangle, \tag{1}$$

where $\Delta^{d+1} := \{\boldsymbol{x} \in \mathbb{R}^{d+1}_{\geq 0} : \boldsymbol{x}^\top \mathbf{1}_{d+1} = 1\}$ represents the $d$-dimensional probability simplex and $\mathbf{A} \in \mathbb{R}^{n \times m}$ is the payoff matrix of the game. Tracing all the way back to Von Neumann's celebrated minimax theorem [von Neumann, 1928], zero-sum games played a pivotal role in the early development of game theory [von Neumann and Morgenstern, 1947] and the crystallization of linear programming duality [Dantzig, 1951]. Indeed, in light of the equivalence between zero-sum games and linear programming [Adler, 2013, von Stengel, 2023, Brooks and Reny, 2023], many central optimization problems can be cast as (1).

38th Conference on Neural Information Processing Systems (NeurIPS 2024).

State of the art algorithms for solving zero-sum games can be coarsely classified based on the desired accuracy of a feasible solution $(\boldsymbol{x}, \boldsymbol{y})$, measured in terms of the *duality gap*

$$\Phi(\boldsymbol{x}, \boldsymbol{y}) \coloneqq \max_{\boldsymbol{y}' \in \Delta^m} \langle \boldsymbol{x}, \mathbf{A}\boldsymbol{y}' \rangle - \min_{\boldsymbol{x}' \in \Delta^n} \langle \boldsymbol{x}', \mathbf{A}\boldsymbol{y} \rangle. \tag{2}$$

In the so-called low-precision regime, where one is content with a crude solution $(\boldsymbol{x}^\star, \boldsymbol{y}^\star)$ such that $\Phi(\boldsymbol{x}^\star, \boldsymbol{y}^\star) =: \epsilon \gg 0$, the best available algorithms typically revolve around the framework of *regret minimization*, both in practice [Farina et al., 2021, Brown and Sandholm, 2019, Zinkevich et al., 2007, Tang et al., 2023] and in theory [Carmon et al., 2020, 2019, 2024, Grigoriadis and Khachiyan, 1995, Clarkson et al., 2012, Alacaoglu and Malitsky, 2022]—in conjunction with other techniques to speed up the per-iteration complexity, such as variance reduction, data structure design, and sparsification [Zhang and Sandholm, 2020, Farina and Sandholm, 2022]. Such algorithms have been central to landmark results in practical computation of equilibrium strategies even in enormous games [Brown and Sandholm, 2018, Bowling et al., 2015, Moravčík et al., 2017, Perolat et al., 2022].

The high-precision regime, where $\epsilon \ll \frac{1}{\mathrm{poly}(nm)}$, has turned out to be more elusive, with current LP-based techniques struggling to scale favorably in large instances. This deficiency can be in part attributed to the relatively high per-iteration complexity of LP-based approaches, such as interior-point methods or the ellipsoid algorithm, as well as their intense memory requirements. A promising antidote is to instead rely on iterative gradient-based methods that have a minimal per-iteration cost. Indeed, in a line of work pioneered by Tseng [1995], it is by known well-documented that *linear convergence*—an iteration complexity scaling only as $\log(1/\epsilon)$—can been achieved even with such methods [Tseng, 1995, Gilpin et al., 2012, Wei et al., 2021, Applegate et al., 2023, Fercoq, 2023]. There is, however, a major caveat to those results: the number of iterations no longer grows polynomially with the dimensions of the game $n$ and $m$, but instead depends on certain condition number-like quantities that could be exponentially large in the description of the problem; it is thus unclear how to interpret those results from a computational standpoint.

To address those shortcomings, in this paper we work in the celebrated framework of *smoothed analysis* pioneered by Spielman and Teng [2004]. Namely, our goal is to characterize the iteration complexity of certain gradient-based algorithms in zero-sum games when the payoff matrix $\mathbf{A}$ is subjected to small but random perturbations, as formally introduced below.

**Definition 1.1** (Zero-sum games under Gaussian perturbations). Let $\bar{\mathbf{A}} \in [-1, 1]^{n \times m}$. We assume that the payoff matrix is given by $\mathbf{A} \coloneqq \bar{\mathbf{A}} + \mathbf{G}$, where each entry of $\mathbf{G}$ is an independent (univariate) Gaussian random variable with zero mean and variance $\sigma^2 \leq 1$.

Randomness here is only injected into the payoff matrix and not the set of constraints (that is, the probability simplex), which is the natural model; after applying the perturbation, the problem should still be a zero-sum game in the form of (1). Under this model, we investigate the convergence of the following gradient-based algorithms.[1] (Their formal description is given later in Appendix B.)

1. *optimistic gradient descent/ascent (`OGDA`)* [Popov, 1980];
2. *optimistic multiplicative weights update (`OMWU`)* [Syrgkanis et al., 2015, Chiang et al., 2012, Rakhlin and Sridharan, 2013];
3. *extra-gradient descent/ascent (`EGDA`)* [Korpelevich, 1976]; and
4. an iterative variant of Nesterov's *smoothing technique (`IterSmooth`)* [Gilpin et al., 2012, Nesterov, 2005].

Smoothed complexity allows interpolating between worst-case analysis—when the variance of the noise $\sigma^2$ is negligible—and average-case analysis—when the noise dominates over the underlying input. An average-case analysis is often unreliable since—as Edelman [1993] convincingly argued—a fully random matrix does not necessarily capture typical instances encountered in practice. Spielman and Teng [2004] put forward the framework of smoothed analysis as an attempt to explain the performance of algorithms in realistic scenarios; to understand how brittle worst-case instances really are. They famously proved that the simplex algorithm, under a certain pivoting rule, enjoys *polynomial smoothed complexity*, meaning that its running time is bounded by some polynomial in the

---

[1]The vanilla gradient descent/ascent algorithm does not even converge (in a last-iterate sense) in zero-sum games (*e.g.*, [Mertikopoulos et al., 2018]), which is why our analysis revolves around certain variants thereof. It is worth noting that regret minimization techniques provide guarantees concerning the average iterates, a distinction blurred in our introduction.

size of the input and $1/\sigma$. Smoothed analysis is by now a well-accepted algorithmic framework with a tremendous impact in the analysis of algorithms. We also argue that it is particularly well-motivated from a game-theoretic perspective: there is often misspecification or noise when modeling a game, so smoothed analysis offers a compelling way of bypassing pathological instances that are perhaps artificial in the first place.

Nevertheless, we are not aware of any prior work operating in the smoothed complexity model per Definition 1.1 in the context of zero-sum games. To clarify this point, it is important to stress here that although zero-sum games can be immediately reduced to linear programs, that reduction is less clear in the smoothed complexity model. In particular, one set of constraints in the induced linear program takes the form $\mathbf{A}\boldsymbol{y} \leq v\mathbf{1}_n =: \boldsymbol{b}$, where $\mathbf{1}_n \in \mathbb{R}^n$ is the all-ones vector. According to the usual model of smoothed complexity in the context of linear programs, randomness has to be injected into both $\mathbf{A}$ and $\boldsymbol{b}$, but that clearly disturbs the validity of the equivalence. More broadly, reductions in the smoothed complexity model are quite delicate [Bläser and Manthey, 2015]; as a further example, even reductions involving solely linear transformations can break in the smoothed complexity model since independence—a crucial assumption in this framework—is not guaranteed to carry over. Relatedly, one interesting direction arising from the work of Spielman and Teng [2003] is to perform smoothed analysis in linear programs which are guaranteed to be feasible and bounded, no matter the perturbation; zero-sum games under Definition 1.1 constitute such a class. Besides the point above, different algorithms designed for the same problem can have entirely different properties, not least in terms of their smoothed complexity. The class of algorithms we consider in this paper is quite distinct from the ones shown to have polynomial smoothed complexity in the context of linear programs (described further in Appendix A). In many ways, gradient-based methods are simpler and more natural, which partly justifies their tremendous practical use. As a result, understanding their smoothed complexity is an important question.

## 1.1 Our results

Our main contribution is to show that, with the exception of `OMWU`, the other gradient-based algorithms mentioned above (Items 1, 3 and 4) have polynomial smoothed complexity with high probability—that is to say, with probability at least $1 - \frac{1}{\text{poly}(nm)}$.

**Theorem 1.2.** *With high probability over the randomness of* $\mathbf{A} \in \mathbb{R}^{n \times m}$ *(Definition 1.1),* `OGDA`, `EGDA` *and* `IterSmooth` *converge to an $\epsilon$-equilibrium after* $\text{poly}(n, m, 1/\sigma) \cdot \log(1/\epsilon)$ *iterations.*

The main takeaway of this result is that, modulo pathological instances, certain gradient-based algorithms are reliable solvers in zero-sum games even in the high-precision regime. Similarly to earlier endeavors in the context of linear programs [Spielman and Teng, 2004, Blum and Dunagan, 2002], a dependency of $\text{poly}(1/\sigma)$ (as in Theorem 1.2) is what we should expect; the one exception is the class of interior-point methods whose running time grows as $\log(1/\sigma)$, but those algorithms are (weakly) polynomial even in the worst case. We further remark that the polynomial dependency on $n$ and $m$ in Theorem 1.2 can almost certainly be improved, and we made no effort to optimize it.

Regarding `OMWU`, which is not covered by Theorem 1.2, we also obtain a significant improvement in the iteration complexity compared to the worst-case analysis of Wei et al. [2021], but our bound is still not polynomial. As we explain further in Appendix C.3, the main difficulty pertaining to `OMWU` is that the analysis of Wei et al. [2021] gives (at best) an exponential bound *no matter the geometry of the problem*. With that mind, our result is essentially the best one could hope for without refining the worst-case analysis of `OMWU`, which is not within our scope here. We anticipate that our characterization herein will prove useful in conjunction with future developments in the worst-case complexity of `OMWU`, as well as in the analysis of other iterative methods.

**The error bound**   The central ingredient that enables gradient-based algorithms to exhibit linear convergence is a certain *error bound*, given below as Definition 1.3. For compactness in our notation, we let $\mathcal{X} := \Delta^n$ and $\mathcal{Y} := \Delta^m$. We then let $\boldsymbol{z} := (\boldsymbol{x}, \boldsymbol{y})$, $\mathcal{Z} := \mathcal{X} \times \mathcal{Y}$, and $\mathcal{Z}^\star := \mathcal{X}^\star \times \mathcal{Y}^\star$, where $\mathcal{X}^\star$ and $\mathcal{Y}^\star$ represent the (convex) set of equilibria for Player $x$ and Player $y$, respectively.

**Definition 1.3** (Error bound). Let $\Phi(\boldsymbol{z})$ denote the duality gap as introduced in (2). We say that the zero-sum game (1) satisfies an *error bound* with modulus $\kappa \in \mathbb{R}_{>0}$ if

$$\Phi(\boldsymbol{z}) \geq \kappa \|\boldsymbol{z} - \Pi_{\mathcal{Z}^\star}(\boldsymbol{z})\| \quad \forall \boldsymbol{z} \in \mathcal{Z}. \tag{3}$$

Above, $\Pi_{\mathcal{Z}^\star}(\cdot)$ denotes the (Euclidean) projection operator; the set of games with a unique equilibrium has measure one, so we can safely replace $\Pi_{\mathcal{Z}^\star}(z)$ by the unique equilibrium $z^\star \in \mathcal{Z}^\star$. It has been known at least since the work of Tseng [1995] that affine variational inequalities indeed satisfy (3). Nevertheless, it should come to no surprise that, even in $3 \times 3$ games, $\kappa$ can be arbitrarily small (Proposition 3.1), which in turn means that, linear convergence notwithstanding, the number of iterations prescribed by an analysis revolving around (3) can be arbitrarily large. In fact, with the exception of OMWU, which is to be discussed further below, Definition 1.3 suffices to establish linear convergence (essentially) based on existing results.[2] Our main result pertaining to Definition 1.3 is that the modulus $\kappa$ is likely to be polynomial in the smoothed complexity model:

**Theorem 1.4.** *With high probability over the randomness of* $\mathbf{A}$ *(Definition 1.1), the error bound per Definition 1.3 is satisfied for any sufficiently small* $\kappa \geq \mathrm{poly}(\sigma, 1/(nm))$.

To establish this result, the first step is to lower bound $\kappa$ in terms of certain natural geometric features of the problem (Theorem 3.6), which is discussed further in Section 3.1. Establishing Theorem 1.4 then reduces to analyzing each of those quantities under Definition 1.1. It turns out that bounding those quantities also suffices for characterizing OMWU, whose existing analysis due to Wei et al. [2021] involves some further ingredients besides the error bound of Definition 1.3.

**Further implications**    Our characterization of the error bound given in Theorem 3.6 has some further important implications. First, a well-known vexing issue regarding computing equilibria even in zero-sum games is that a solution with small duality gap can still be relatively far from the equilibrium in the geometric sense, a phenomenon further exacerbated in multi-player games [Etessami and Yannakakis, 2007]. Therefore, results providing guarantees in terms of the duality gap are not particularly informative when it comes to computing strategies close to the equilibrium in a geometric sense. At the same time, there are ample reasons why the latter guarantee is more appealing [Etessami and Yannakakis, 2007]. Theorem 1.4 implies that such concerns can be alleviated in the smoothed complexity model:

**Corollary 1.5.** *With high probability over the randomness of* $\mathbf{A}$ *(Definition 1.1), any point* $z \in \mathcal{Z}$ *with* $\Phi(z) \leq \epsilon$ *satisfies* $\|z - z^\star\| \leq \epsilon \cdot \mathrm{poly}(n, m, 1/\sigma)$.

Beyond smoothed analysis, Theorem 3.6 applies to any non-degenerate game (Definition 3.2), and can be thereby used to parameterize the rate of convergence of gradient-based algorithms based on natural and interpretable game-theoretic quantities of the underlying game, which has eluded prior work. In particular, we make a natural connection between the complexity of gradient-based algorithms and *perturbation stability* properties of the equilibrium. In light of misspecifications which are often present in game-theoretic modeling, focusing on games with perturbation-stable equilibria is well-motivated and has already received ample of interest in prior work [Balcan and Braverman, 2017, Awasthi et al., 2010]; more broadly, perturbation stability is a common assumption in the analysis of algorithms beyond the worst-case model [Makarychev and Makarychev, 2021]. There are different natural ways of defining perturbation-stable games; here, we assume that any perturbation with magnitude below $\delta > 0$, in that $\|\mathbf{A}' - \mathbf{A}\|_2 \leq \delta$, maintains the support of the equilibrium and the non-degeneracy of the game; we call such games $\delta$-*support-stable* (Definition 4.1). In this context, we show the following result.

**Corollary 1.6.** *For any* $\delta$-*support-stable zero-sum game,* OGDA, EGDA *and* IterSmooth *converge to an* $\epsilon$-*equilibrium after* $\mathrm{poly}(n, m, 1/\delta) \cdot \log(1/\epsilon)$ *iterations.*

That is, games in which $\delta$ is not too close to $0$ are more amenable to gradient-based algorithms, which is a quite natural connection. Corollary 1.6 is shown by relating each of the quantities involved in Theorem 3.6 to parameter $\delta$ defined above.

## 2    Notation

Before we proceed with our technical content, we first take the opportunity to streamline our notation; further background on smoothed analysis and a description of the algorithms referred to earlier (Items 1 to 4) is given later in Appendix B, as it is not important for the purpose of the main body.

---

[2]Definition 1.3 also readily establishes linear convergence for other compelling primal-dual algorithms, as shown recently by Applegate et al. [2023]; in that paper, the error bound was referred to as "sharpness," a terminology employed in other papers as well (*e.g.*, [Zarifis et al., 2024]).

We use boldface letters, such as $\boldsymbol{x}, \boldsymbol{y}, \boldsymbol{b}, \boldsymbol{c}$, to represent vectors in a Euclidean space. For a vector $\boldsymbol{x} \in \mathbb{R}^n$, we access its $i$th coordinate via a subscript, namely $\boldsymbol{x}_i$. Superscripts (together with parantheses) are typically reserved for the (discrete) time index. We denote by $\|\boldsymbol{x}\|$ the Euclidean norm, $\|\boldsymbol{x}\| := \sqrt{\sum_{i=1}^n \boldsymbol{x}_i^2}$, the $\ell_\infty$ norm by $\|\boldsymbol{x}\|_\infty := \max_{1 \le i \le n} |\boldsymbol{x}_i|$, and the $\ell_1$ norm by $\|\boldsymbol{x}\|_1 := \sum_{i=1}^n |\boldsymbol{x}_i|$. For $\boldsymbol{x}, \boldsymbol{x}' \in \mathbb{R}^n$, we let $\mathsf{dist}(\boldsymbol{x}, \boldsymbol{x}') := \|\boldsymbol{x} - \boldsymbol{x}'\|$. $\mathsf{span}(\cdot)$ represents the linear space spanned by a given set of vectors. For $\boldsymbol{x} \in \mathbb{R}^n$ and a subset $B \subseteq [n]$, we denote by $\boldsymbol{x}_B \in \mathbb{R}^B$ the subvector of $\boldsymbol{x}$ induced by $B$. We let $\mathbf{1}_n \in \mathbb{R}^n$ be the all-ones vector of dimension $n$; we will typically omit the subscript when it is clear from the context. For vectors $\boldsymbol{x} \in \mathbb{R}^n$ and $\boldsymbol{y} \in \mathbb{R}^m$, we write $(\boldsymbol{x}, \boldsymbol{y}) \in \mathbb{R}^{n+m}$ to denote their concatenation. Throughout this paper, we use $\boldsymbol{x}$ and $\boldsymbol{y}$ to denote the strategy of Player $x$ and Player $y$, respectively.

To represent matrices, we use boldface capital letter, such as $\mathbf{A}, \mathbf{Q}$. It will sometimes be convenient to use $\mathbf{A}^\flat \in \mathbb{R}^{nm}$ to represent a vectorization of $\mathbf{A} \in \mathbb{R}^{n \times m}$. We overload notation by letting $\|\mathbf{A}\|$ be the spectral norm of $\mathbf{A}$. For a matrix $\mathbf{A} \in \mathbb{R}^{n \times m}$ and subsets $B \subseteq [n], N \subseteq [m]$, we denote by $\mathbf{A}_{B,N} \in \mathbb{R}^{B \times N}$ the submatrix of $\mathbf{A}$ induced by $B$ and $N$. $\mathbf{A}_{i,:}$ and $\mathbf{A}_{:,j}$ represent the $i$th row and $j$th column of $\mathbf{A}$, respectively. The singular values of a matrix $\mathbf{M} \in \mathbb{R}^{d \times d}$ are denoted by $\sigma_1(\mathbf{M}) \ge \sigma_2(\mathbf{M}) \ge \cdots \ge \sigma_d(\mathbf{M}) \ge 0$ (not to be confused with our notation for the variance $\sigma^2$). To be more explicit, we may also use $\sigma_{\max}(\mathbf{M}) := \sigma_1(\mathbf{M})$ and $\sigma_{\min}(\mathbf{M}) := \sigma_d(\mathbf{M})$.

# 3 Smoothed analysis of the error bound

In this section, we perform a smoothed analysis of the error bound—as introduced earlier in Definition 1.3—in (two-player) zero-sum games. It is first instructive to point out why smoothed analysis is useful in the first place: the modulus $\kappa$ can be arbitrarily close to $0$ even when $n = m = 3$ (that is, $3 \times 3$ games); this is detrimental as the iteration complexity of algorithms such as OGDA grows as a polynomial in $1/\kappa$.

**Proposition 3.1.** *There exists a $3 \times 3$ zero-sum game such that $\kappa$ per Definition 1.3 is arbitrarily close to $0$.*

In proof, it is enough to consider the ill-conditioned diagonal matrix

$$\mathbf{A} = \begin{pmatrix} \gamma & 0 & 0 \\ 0 & 2\gamma & 0 \\ 0 & 0 & 1 \end{pmatrix}, \tag{4}$$

where $0 < \gamma \ll 1$. The (unique) equilibrium of (4) reads $\boldsymbol{x}^\star = \boldsymbol{y}^\star = \frac{1}{3+2\gamma}(2, 1, 2\gamma) \in \Delta^3$. Now, considering $\boldsymbol{x} = (1, 0, 0)$ and $\boldsymbol{y} = (0, 0, 1)$, for the duality gap we have $\Phi(\boldsymbol{x}, \boldsymbol{y}) = \gamma$, while the distance of $(\boldsymbol{x}, \boldsymbol{y})$ from the optimal solution $(\boldsymbol{x}^\star, \boldsymbol{y}^\star)$ is at least $\frac{3}{3+2\gamma}$. In turn, by Definition 1.3, this means that $\kappa \le 2\gamma$. So, Proposition 3.1 follows by taking $\gamma \to 0$.[3]

Proposition 3.1 exposes one type of pathology that can decelerate gradient-based algorithms, which is evidently related to the poor spectral properties of the payoff matrix. This intuition is quite helpful when equilibria are fully supported—as is the case in (4)—but has to be significantly refined more broadly, as we formalize in the sequel.

To sidestep such pathological examples, we thus turn to the smoothed analysis framework of Definition 1.1.

## 3.1 Overview

The most natural approach to analyze the error bound in the smoothed complexity model is to rely on an existing (worst-case) analysis proving that a positive $\kappa$ exists, and then attempt to refine that analysis. Yet, at least based on such prior results we are aware of, that turns out to be challenging. As an example, let us consider the recent analysis of Wei et al. [2021]. As we explain in more detail in Appendix C.3, Wei et al. [2021] relate the modulus $\kappa$ of the error bound to the (inverse of the) norm of a solution to a certain feasible linear program; the existence of a legitimate $\kappa > 0$ then follows readily from feasibility. Now, this reduction seems quite promising: Renegar [1994] has shown that the

---

[3]If we want to specify the game with a (finite) number of $L$ bits, Proposition 3.1 tells us that the modulus $\kappa$ can be exponentially small in $L$.

norm of a solution to a linear program can be bounded in terms of its *condition number*—the distance to infeasibility in our case, and Dunagan et al. [2011] later proved that the condition number of linear programs is polynomial in the smoothed complexity model. Nevertheless, there are some difficulties in materializing that argument. First, the induced linear program involves terms depending on both the payoff matrix and the geometry of the constraints (the probability simplex in our case). Consequently, the analysis of Dunagan et al. [2011] does not carry over since randomness is only injected into the payoff matrix. The second and more important obstacle is that the induced linear program depends on the optimal solution, which in turn depends on the randomness of the payoff matrix; this significantly entangles the underlying distribution. As there are exponentially many possible configurations, we cannot afford to argue about each one separately and then apply the union bound. This difficulty is in fact known to be the crux in performing smoothed analysis [Spielman and Teng, 2004].[4]

To address those challenges, we provide a new characterization of the error bound in terms of some natural quantities of the underlying game (Theorem 3.6), which in some sense capture the difficulty of the problem. We are then able to use a technique due to Spielman and Teng [2004], exposed in Section 3.3, to bound the probability that each of the involved quantities is close to $0$ (Propositions 3.8 to 3.10), even though the underlying distribution is quite convoluted. The resulting analysis follows the one given by Spielman and Teng [2003] in the context of termination of linear programs, but still has to account for a number of structural differences.

In what follows, we structure our argument as follows. First, in Section 3.2, we relate the modulus $\kappa$ to some natural quantities capturing key geometric features of the problem. Section 3.3 then proceed by analyzing those quantities in the smoothed analysis framework.

## 3.2 Characterization of the error bound

Our first goal is to characterize the error bound in terms of certain natural quantities, which will then enable us to provide polynomial error bounds in the smoothed complexity model. Our only assumption here is that the zero-sum game is *non-degenerate*, in the sense of Definition 3.2 below; this can always be met with the addition of an arbitrarily small amount of noise (Lemma C.1). As such, our characterization here has an interest beyond the smoothed analysis framework, casting the error bound in terms of more interpretable game-theoretic quantities; for example, a concrete implication is given in Section 4.

Let us denote by $v$ the *value* of game (1), that is,

$$v = \min_{\boldsymbol{x} \in \mathcal{X}} \max_{\boldsymbol{y} \in \mathcal{Y}} \langle \boldsymbol{x}, \mathbf{A}\boldsymbol{y} \rangle = \max_{\boldsymbol{y} \in \mathcal{Y}} \min_{\boldsymbol{x} \in \mathcal{X}} \langle \boldsymbol{x}, \mathbf{A}\boldsymbol{y} \rangle,$$

which is a consequence of the minimax theorem [von Neumann, 1928]. We are now ready to state the formal definition of a non-degenerate game.

**Definition 3.2** (Non-degenerate game)**.** A zero-sum game described with a payoff matrix $\mathbf{A}$ and value $v$ is said to be *non-degenerate* if it admits a unique equilibrium $(\boldsymbol{x}^\star, \boldsymbol{y}^\star) \in \mathcal{Z}$, and $\boldsymbol{x}^\star$ and $\boldsymbol{y}^\star$ make tight exactly $n$ of the inequalities $\{\boldsymbol{x}_i \geq 0\}_{i \in [n]} \cup \{\langle \boldsymbol{x}, \mathbf{A}_{:,j} \rangle \leq v\}_{j \in [m]}$ and $m$ of the inequalities $\{\boldsymbol{y}_j \geq 0\}_{j \in [m]} \cup \{\langle \boldsymbol{y}, \mathbf{A}_{i,:} \rangle \geq v\}_{i \in [n]}$, respectively.

In the sequel, we will make constant use of the fact that the set of degenerate games has measure zero under the law induced by Definition 1.1 (Lemma C.1).

In this context, we let $B(\boldsymbol{x}^\star) := \{i \in [n] : \boldsymbol{x}_i^\star > 0\}$ denote the *support* of $\boldsymbol{x}^\star$ (corresponding to Player $x$), and similarly $N(\boldsymbol{y}^\star) := \{j \in [m] : \boldsymbol{y}_j^\star > 0\}$ for the support of Player $y$. The strict complementarity theorem [Ye, 2011] tells us that $B$ indexes exactly the set of tight inequalities $\{\langle \boldsymbol{y}, \mathbf{A}_{i,:} \rangle \geq v\}_{i \in [n]}$, and symmetrically, $N$ indexes exactly the set of tight inequalities $\{\langle \boldsymbol{x}, \mathbf{A}_{:,j} \rangle \leq v\}_{j \in [m]}$. In particular, this implies that $|B| = |N|$ with probability 1. It will also be convenient to define $\overline{B} := [n] \setminus B$ and $\overline{N} := [m] \setminus N$.

Now, at a high level, one can split solving a zero-sum game into two subproblems: i) identifying the support of the equilibrium, and ii) solving the induced *linear system* to specify the exact probabilities

---

[4] This is not a concern in the *unconstrained* setting, where $\mathcal{X} = \mathbb{R}^n$ and $\mathcal{Y} = \mathbb{R}^m$, in which a polynomial smoothed complexity follows readily from existing results relating the convergence of `OGDA` or `EGDA` to the condition number of the payoff matrix $\mathbf{A}$ (*e.g.*, [Mokhtari et al., 2020, Li et al., 2023, Azizian et al., 2020]), which in turn is well-known to be polynomial in the smoothed complexity model [Spielman and Teng, 2004].

within the support. It will be helpful to have that viewpoint in mind in the upcoming analysis, and in particular in the proof of Theorem 3.6. Roughly speaking, thinking of $\kappa$ as a measure of the problem's difficulty, we will relate $\kappa$ to i) the difficulty of identifying the support of the equilibrium, and ii) the difficulty of solving the induced linear system. To be clear, those two subproblems are only helpful for the purpose of the analysis, and they are certainty intertwined when using algorithms such as OGDA.

Staying on the latter task, we will make use of a certain transformation so as to eliminate one of the redundant variables. Namely, for any $\widehat{\boldsymbol{x}}_B \in \Delta(B)$ and $\widehat{\boldsymbol{y}}_N \in \Delta(N)$, let us select a fixed pair of coordinates $(i, j) \in B \times N$ (for example, the ones with the smallest index). Using the fact that $\langle \widehat{\boldsymbol{x}}_B, \mathbf{1} \rangle = 1$ and $\langle \widehat{\boldsymbol{y}}_N, \mathbf{1} \rangle = 1$, we can eliminate $\widehat{\boldsymbol{x}}_i$ and $\widehat{\boldsymbol{y}}_j$ by writing

$$\langle \widehat{\boldsymbol{x}}_B, \mathbf{A}_{B,N} \widehat{\boldsymbol{y}}_N \rangle = \langle \widetilde{\boldsymbol{x}}, \mathbf{Q} \widetilde{\boldsymbol{y}} \rangle - \langle \widetilde{\boldsymbol{x}}, \boldsymbol{c} \rangle - \langle \widetilde{\boldsymbol{y}}, \boldsymbol{b} \rangle + d, \tag{5}$$

where $\widetilde{\boldsymbol{x}} \in \mathbb{R}^{\widetilde{B}}_{\geq 0}, \widetilde{\boldsymbol{y}} \in \mathbb{R}^{\widetilde{N}}_{\geq 0}$ (for $\widetilde{B} := B \setminus \{i\}$ and $\widetilde{N} := N \setminus \{j\}$) coincide with $\widehat{\boldsymbol{x}}_B$ and $\widehat{\boldsymbol{y}}_N$ on all coordinates in $\widetilde{B}$ and $\widetilde{N}$, respectively, and $\mathbf{A}^{\flat}_{B,N} = \mathbf{T}(\mathbf{Q}^{\flat}, \boldsymbol{b}, \boldsymbol{c}, d)$ for a (non-singular) linear transformation $\mathbf{T} \in \mathbb{R}^{(BN) \times (BN)}$. (We spell out the exact definition of $\mathbf{T}$ later in Appendix C.1, as it is not important for our purposes here; it follows by simply writing $\widehat{\boldsymbol{x}}_i = 1 - \langle \widetilde{\boldsymbol{x}}, \mathbf{1} \rangle$ and $\widehat{\boldsymbol{y}}_j = 1 - \langle \widetilde{\boldsymbol{y}}, \mathbf{1} \rangle$.) The point of transformation (5) is that, by eliminating one of the redundant variables, there is a convenient characterization of the equilibrium (Claim C.3); namely, $\mathbf{Q} \boldsymbol{y}^{\star} = \boldsymbol{c}$ and $\mathbf{Q}^{\top} \boldsymbol{x}^{\star} = \boldsymbol{b}$.

We are now ready to introduce the key quantities upon which our characterization relies on. It turns out that those are analogous to the ones considered by Spielman and Teng [2003] in the context of analyzing the termination of linear programs; this is not coincidental, as our analysis was especially targeted to do so.

**Definition 3.3.** Let $\mathbf{A}$ be the payoff matrix of a non-degenerate game, $(\boldsymbol{x}^{\star}, \boldsymbol{y}^{\star}) \in \mathcal{Z}$ the unique equilibrium, and $B \subseteq [n], N \subseteq [m]$ the support of $\boldsymbol{x}^{\star}$ and $\boldsymbol{y}^{\star}$ respectively. We introduce the following quantities.

1. $\alpha_P(\mathbf{A}) := \min_{i \in B}(\boldsymbol{x}^{\star}_i)$ and $\alpha_D(\mathbf{A}) := \min_{j \in N}(\boldsymbol{y}^{\star}_j)$;

2. $\beta_P(\mathbf{A}) := \min_{j \in \overline{N}}(v - \langle \boldsymbol{x}^{\star}_B, \mathbf{A}_{B,j} \rangle)$ and $\beta_D(\mathbf{A}) := \min_{i \in \overline{B}}(\langle \mathbf{A}_{i,N}, \boldsymbol{y}^{\star}_N \rangle - v)$; and

3. $\gamma_P(\mathbf{A}) := \min_j \mathsf{dist}(\mathbf{Q}_{:,j}, \mathsf{span}(\mathbf{Q}_{:,\widetilde{N}-j}))$ and $\gamma_D(\mathbf{A}) := \min_i \mathsf{dist}(\mathbf{Q}_{i,:}, \mathsf{span}(\mathbf{Q}_{\widetilde{B}-i,:}))$, where we use the shorthand notation $\widetilde{B} - i := \widetilde{B} \setminus \{i\}$ ($\widetilde{N} - j := \widetilde{N} \setminus \{j\}$), and $\mathbf{Q} = \mathbf{Q}(\mathbf{A})$ is defined in (5).

(Above, we adopt the convention that if a minimization problem is with respect to an empty set, the minimum is to be evaluated as 1.)

Item 3 above will enable us to control the norm of solutions to any linear system induced by $\mathbf{Q}$, as we explain in the sequel. Our proof will actually rely on a slightly different matrix, which we call $\overline{\mathbf{Q}}$; the lemma below relates the geometry of $\overline{\mathbf{Q}}$ to $\mathbf{Q}$, and reassures us that the condition number of $\overline{\mathbf{Q}}$ cannot be far from that of $\mathbf{Q}$ so long as $1 - \sum_{j \in \widetilde{N}} \boldsymbol{y}^{\star}_j \geq \alpha_D(\mathbf{A})$ (by Item 1) is not too close to 0. (A symmetric statement holds when focusing on Player $y$.)

**Lemma 3.4.** Let $\boldsymbol{c} = \mathbf{Q} \widetilde{\boldsymbol{y}}^{\star} = \sum_{j \in \widetilde{N}} \widetilde{\boldsymbol{y}}^{\star}_j \mathbf{Q}_{:,j}$, and suppose that $\overline{\mathbf{Q}} \in \mathbb{R}^{\widetilde{B} \times \widetilde{N}}$ is such that its $j$th column is equal to $\mathbf{Q}_{:,j} - \boldsymbol{c}$. Then,

$$\min_{j \in \widetilde{N}} \mathsf{dist}(\mathbf{Q}_{:,j}, \mathsf{span}(\mathbf{Q}_{:,\widetilde{N}-j})) \leq \left( 1 + \frac{|\widetilde{N}|}{1 - \sum_{j \in \widetilde{N}} \boldsymbol{y}^{\star}_j} \right) \min_{j \in \widetilde{N}} \mathsf{dist}(\overline{\mathbf{Q}}_{:,j}, \mathsf{span}(\overline{\mathbf{Q}}_{:,\widetilde{N}-j})).$$

Next, we recall a fairly standard bound relating the magnitude of a solution to a linear system $\widetilde{\boldsymbol{x}} = \mathbf{M} \boldsymbol{p}$ with the smallest singular value of a full-rank matrix $\mathbf{M}$.

**Lemma 3.5.** Let $\mathbf{M} \in \mathbb{R}^{d \times d}$ be a full-rank matrix. For any $\widetilde{\boldsymbol{x}} \in \mathbb{R}^d$ there is $\boldsymbol{p} \in \mathbb{R}^d$ with $\|\boldsymbol{p}\| \leq \frac{1}{\sigma_{\min}(\mathbf{M})} \|\widetilde{\boldsymbol{x}}\|$ such that

$$\widetilde{\boldsymbol{x}} = \mathbf{M} \boldsymbol{p} = \sum_{j=1}^{d} \boldsymbol{p}_j \mathbf{M}_{:,j}.$$

Moreover, to connect Lemma 3.5 with $\gamma_P(\mathbf{A})$, we observe that the smallest singular value can also be lower bounded in terms of the smallest distance of a column from the linear space spanned by the rest of the columns—which now matches the expression of Item 3 we saw earlier. In particular, we will make use of the so-called negative second moment identity [Tao et al., 2010] (Proposition C.4), which implies that

$$\sigma_{\min}(\overline{\mathbf{Q}}) \geq \sqrt{\frac{1}{\sum_{j \in \widetilde{N}} \mathsf{dist}^{-2}(\overline{\mathbf{Q}}_{:,j}, \mathsf{span}(\overline{\mathbf{Q}}_{:,\widetilde{N}-j}))}} \geq \frac{1}{\sqrt{|\widetilde{N}|}} \min_{j \in \widetilde{N}} \mathsf{dist}(\overline{\mathbf{Q}}_{:,j}, \mathsf{span}(\overline{\mathbf{Q}}_{:,\widetilde{N}-j})). \quad (6)$$

Proposition C.4 also implies that $\gamma_D(\mathbf{A}) \geq \frac{1}{\sqrt{|\overline{B}|}} \gamma_P(\mathbf{A})$, and so it will suffice to lower bound $\gamma_P(\mathbf{A})$ in the sequel. We are now ready to proceed with the main result of this subsection. Below, we use the notation "$\gtrsim$" to suppress lower-order terms and absolute constants.

**Theorem 3.6.** *Let $\mathbf{A}$ be a non-degenerate payoff matrix, and suppose that $(\alpha_P(\mathbf{A}), \alpha_D(\mathbf{A}))$, $(\beta_P(\mathbf{A}), \beta_D(\mathbf{A}))$ and $(\gamma_P(\mathbf{A}), \gamma_D(\mathbf{A}))$ are as in Definition 3.3. Then, the error bound (Definition 1.3) is satisfied for any sufficiently small modulus*

$$\kappa \gtrsim \frac{1}{\|\mathbf{A}^\flat\|_\infty} \frac{1}{\min(n,m)^3} \min \left\{ (\alpha_D(\mathbf{A}))^2 \beta_D(\mathbf{A}) \gamma_P(\mathbf{A}), (\alpha_P(\mathbf{A}))^2 \beta_P(\mathbf{A}) \gamma_D(\mathbf{A}) \right\}.$$

It is enough to explain how to lower bound $\kappa > 0$ such that $\max_{\mathbf{y}' \in \mathcal{Y}} \langle \mathbf{x}, \mathbf{A}\mathbf{y}' \rangle - v \geq \kappa \|\mathbf{x} - \Pi_{\mathcal{X}^\star}(\mathbf{x})\| = \kappa \|\mathbf{x} - \mathbf{x}^\star\|$ for any $\mathbf{x} \in \mathcal{X}$. In a nutshell, our argument is divided based on the magnitude $\lambda := \|\mathbf{x}_B\|$, which can be thought of as a measure of closeness from the support of the equilibrium. When $\lambda \ll 1$, which means that $\mathbf{x}$ is still far from the support of the equilibrium, $\max_{\mathbf{y}' \in \mathcal{Y}} \langle \mathbf{x}, \mathbf{A}\mathbf{y}' \rangle - v$ is governed by $\beta_D(\mathbf{A})$. In the contrary case, our basic strategy revolves around showing that the error bound can be treated as in the unconstrained case, which would then relate the modulus $\kappa$ to the smallest singular value of the underlying matrix (essentially by Lemma 3.5)—and subsequently to $\gamma_P(\mathbf{A})$ due to (6). Indeed, this turns out to be possible by working with matrix $\overline{\mathbf{Q}}$, as defined earlier in Lemma 3.4. We defer the precise argument to Appendix C.1.

### 3.3 Smoothed analysis

Having established Theorem 3.6, our next step is to show that each of the quantities introduced in Definition 3.3 is unlikely to be too close to $0$ in the smoothed complexity model, which would then imply Theorem 1.4. The main difficulty lies in the fact that each configuration that may arise depends on the support of the equilibrium, which in turn depends on the underlying randomization of $\mathbf{A}$, thereby significantly complicating the underlying distribution. Further, one cannot afford to argue about each configuration separately and then apply the union bound as there are too many possible configurations. To tackle this challenge, we follow the approach put forward by Spielman and Teng [2003].

In particular, given that all quantities of interest in Theorem 3.6 depend on the support of the equilibrium, it is natural to proceed by partitioning the probability space over all possible supports, and then bound the worst possible one—that is, the one maximizing the probability we want to minimize. In doing so, the challenge is that one has to condition on the equilibrium having a given support (formally justified by Proposition C.5). To argue about the induced probability density function upon such a conditioning, it is convenient to perform a change of variables from $\mathbf{A}$ to a new set of variables that now contains the equilibrium $(\mathbf{x}^\star, \mathbf{y}^\star)$ (Lemma C.6). The basic idea here is that since the event we condition on concerns the equilibrium, it is helpful to have that equilibrium being part of our set of variables. The induced probability density function is now quite complicated, but can still be analyzed using the following lemma.

**Lemma 3.7** (Spielman and Teng, 2003). *Let $\rho$ be the probability density function of a random variable $X$. If there exist $\delta > 0$ and $c \in (0, 1]$ such that*

$$0 \leq t \leq t' \leq \delta \implies \frac{\rho(t')}{\rho(t)} \geq c, \quad (7)$$

*then*

$$\mathbb{P}[X \leq \epsilon \mid X \geq 0] \leq \frac{\epsilon}{c\delta}.$$

In words, random variables whose density is smooth—in the sense of (7)—are unlikely to be too close to 0. Gaussian random variables certainly have that property (Lemma C.8), but it is not confined to the Gaussian law; the analysis of Spielman and Teng [2003]—and subsequently our result—is not tailored to the Gaussian case.

We are now ready to state our main results in the smoothed complexity model; the proofs are deferred to Appendix C.2. We commence with $\beta_P(\mathbf{A})$, which is the easiest to analyze. In particular, the following result is a consequence of an anti-concentration bound with respect to a conditional Gaussian random variable (Lemma C.7).

**Proposition 3.8.** *Let* $\beta_P(\mathbf{A})$ *be defined as in Item 2. For any* $\epsilon \geq 0$,

$$\mathbb{P}_{\mathbf{A}} \left[ \beta_P(\mathbf{A}) \leq \frac{\epsilon}{5\|\mathbf{A}^\flat\|_\infty} \right] \leq \epsilon \frac{e \min(n,m)^2}{\sigma^2}.$$

The analysis of $\gamma_P(\mathbf{A})$ is more challenging, and makes crucial use of Lemma 3.7. As we alluded to earlier, a key step is to change variables from $\mathbf{A}_{B,N}$ to $(\mathbf{Q}, \boldsymbol{b}, \boldsymbol{c}, \cdot)$—in accordance with (5)—and then to $(\mathbf{Q}, \boldsymbol{x}^\star, \boldsymbol{y}^\star, \cdot)$ based on $\mathbf{Q}\widetilde{\boldsymbol{y}}^\star = \boldsymbol{c}$, $\mathbf{Q}^\top \widetilde{\boldsymbol{x}}^\star = \boldsymbol{b}$. It is important to note that $\mathbf{Q}$ no longer contains independent random variables even though $\mathbf{A}_{B,N}$ is (by Definition 1.1); this stems from the presence of a redundant variable in $\boldsymbol{x}_B^\star$ (since $\langle \boldsymbol{x}_B^\star, \mathbf{1} \rangle = 1$). Nevertheless, we can still overcome this issue using Lemma 3.7, leading to the following bound.

**Proposition 3.9.** *Let* $\gamma_P(\mathbf{A})$ *be defined as in Item 3. For any* $\epsilon \geq 0$,

$$\mathbb{P}_{\mathbf{A}} \left[ \gamma_P(\mathbf{A}) \leq \frac{\epsilon}{4 \max_{j \in \widetilde{N}} \|\mathbf{Q}_{:,j}\| + 20\|\mathbf{A}^\flat\|_\infty + 3} \right] \leq \epsilon \frac{4e \min(n,m)^3}{\sigma^2}.$$

Similar reasoning, albeit with some further complications, provides a bound for $\alpha_P(\mathbf{A})$, which is given below.

**Proposition 3.10.** *Let* $\alpha_P(\mathbf{A})$ *be defined as in Item 1. For any* $\epsilon \geq 0$,

$$\mathbb{P}_{\mathbf{A}} \left[ \alpha_P(\mathbf{A}) \leq \frac{\epsilon}{25(\|\mathbf{A}^\flat\|_\infty + 1)^2} \right] \leq \epsilon \frac{8e^2 mn \min(n,m)}{\sigma^2}.$$

Armed with Propositions 3.8 to 3.10 and Theorem 3.6, we can establish Theorem 1.2 by suitably leveraging existing results, as we formalize in Appendix C.3.

# 4 Parameterized results for perturbation-stable games

Another important implication of our characterization in Theorem 3.6 is that it enables connecting the convergence rate of gradient-based algorithms to natural and interpretable game-theoretic quantities. In particular, here we highlight a connection with perturbation-stable games, in the following formal sense.

**Definition 4.1** (Perturbation-stable games)**.** Let $\mathbf{A}$ be the payoff matrix of a non-degenerate game. We say that the game is $\delta$-*support-stable*, with $\delta > 0$, if for any $\mathbf{A}'$ with $\|\mathbf{A} - \mathbf{A}'\| \leq \delta$ it holds that $\mathbf{A}'$ is a non-degenerate game whose equilibrium has the same support as $\mathbf{A}$.

Perhaps the simplest example of a support-stable game with a favorable parameter $\delta > 0$ arises when $\mathbf{A}$ is the $2 \times 2$ identity matrix. Indeed, as long as the perturbation parameter $\delta$ remains below a certain absolute constant, the perturbed game still admits a unique full-support equilibrium. To see this, suppose for the sake of contradiction that the perturbed game has an equilibrium such that Player $x$ plays one of the two actions with probability $1$. Player $y$ would then obtain a utility of at least $1 - O(\delta)$. But the value of the original game was $1/2$, which in turn implies that the value of the perturbed game is $1/2 \pm \Theta(\delta)$; for a sufficiently small $\delta$ this leads to a contradiction. Similar reasoning applies with respect to Player $y$. (The previous argument carries over more broadly to diagonally dominant $2 \times 2$ payoff matrices.)

As we have highlighted already, games with perturbation-stable equilibria—albeit under different notions of stability—have already received attention in the literature [Balcan and Braverman, 2017, Awasthi et al., 2010] (*cf.* Cohen [1986]), and are part of a broader trend in the analysis of algorithms beyond the worst case (for further background, we refer to the excellent book edited by Roughgarden [2021]). Our goal here is to make the following natural connection.

**Theorem 4.2.** *Any $\delta$-support-stable game (per [Definition 4.1](#)) satisfies the error bound for any sufficiently small modulus*

$$\kappa \geq \mathsf{poly}\left(\frac{1}{n}, \frac{1}{m}, \delta\right).$$

By virtue of our discussion in [Appendix C.3](#), [Theorem 4.2](#) immediately implies [Corollary 1.6](#). Indeed, we observe that all parameters involved in [Theorem 3.6](#) can be lower bounded in terms of the stability parameter of [Definition 4.1](#), as we formalize in [Appendix C.4](#).

## 5 Conclusions and future research

In conclusion, we performed the first smoothed analysis with respect to a number of well-studied gradient-based algorithms in zero-sum games. In particular, we showed that `OGDA`, `EGDA` and `IterSmooth` all enjoy polynomial smoothed complexity, meaning that their iteration complexity grows as a polynomial in the dimensions of the game, $1/\sigma$, and $\log(1/\epsilon)$; for `OMWU`, our analysis reveals a significant improvement over the worst-case bound due to Wei et al. [2021], but it still remains superpolynomial. We also made a connection between the rate of convergence of the above algorithms and a natural perturbation-stability property of the equilibrium, which is interesting beyond the model of smoothed complexity.

A number of interesting avenues for future research remain open. First, is it the case that `OMWU` has polynomial smoothed complexity or is there an inherent separation with the other algorithms we studied? Answering this question in the positive would necessitate significantly improving the worst-case analysis of `OMWU` due to Wei et al. [2021] (*cf.* Cai et al. [2024] for a recent development concerning the last-iterate convergence of `OMWU`). Beyond `OMWU`, our results could also prove useful for establishing polynomial bounds for other natural dynamics in the smoothed analysis framework. Moreover, our characterization of the error bound in [Theorem 3.6](#) assumes that the game is non-degenerate. This is an innocuous assumption in the smoothed complexity model, as it holds with probability 1, but nevertheless it would be interesting to generalize it to any game. Doing so could shed some light into whether [Theorem 4.2](#) holds with respect to other, perhaps more natural notions of perturbation stability beyond [Definition 4.1](#). It would also be interesting to investigate other models of smoothed complexity that account for dependencies between the entries of the payoff matrix [Bhaskara et al., 2024]. Moreover, our focus has been on zero-sum games under simplex constraints, but we suspect that more general positive results should be attainable under polyhedral constraint sets; perhaps the most notable such candidate is the class of *extensive-form games* [Romanovskii, 1962, von Stengel, 1996]. Even beyond (two-player) zero-sum games, [Theorem 1.2](#) could apply to (multi-player) *polymatrix* zero-sum games [Cai et al., 2016]. It is less clear whether the model of smoothed complexity can be informative when it comes to convergence to *coarse correlated equilibria* in multi-player games.

## Acknowledgments

We are grateful to the anonymous reviewers at NeurIPS for their helpful feedback. The first author is indebted to Ioannis Panageas for many insightful discussions. This material is based on work supported by the Vannevar Bush Faculty Fellowship ONR N00014-23-1-2876, National Science Foundation grants RI-2312342 and RI-1901403, ARO award W911NF2210266, and NIH award A240108S001.

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

# A Further related work

Besides the pioneering work of Spielman and Teng [2004], which revolved around the simplex algorithm, other prominent algorithms for solving linear programs have also been investigated through the lens of smoothed complexity. Blum and Dunagan [2002] showed that perceptron, a popular algorithm in machine learning, also enjoys a polynomial smoothed complexity (with high probability) for solving linear programming feasibility problems, which can also capture general linear programs via a binary search procedure. Further, Dunagan et al. [2011] performed a smoothed analysis of interior-point methods by relying on an earlier characterization due to Renegar [1995].

Beyond linear programming and (two-player) zero-sum games, there has been a considerable interest in understanding the smoothed complexity of Nash equilibria in general-sum games, but the outlook that has emerged from this endeavor is rather bleak [Chen et al., 2009, Boodaghians et al., 2020, Rubinstein, 2016]. On a more positive note, Daskalakis et al. [2024] recently considered a more permissive solution concept they refer to as a *smooth Nash equilibrium*; the basic idea of their relaxation is that instead of considering best-response deviations, they restrict to deviations that do not assign too much probability mass on any pure strategy, as controlled by a certain parameter. For a certain regime of that parameter, they obtained positive results, bypassing the intractability of the usual Nash equilibrium. Considering smooth Nash equilibria could also be fruitful in the context of zero-sum games. In particular, we surmise that, if one is content with convergence to smooth Nash equilibria, the error bound could exhibit more favorable properties. Smoothed analysis has also been applied to more structured classes of games, such as congestion or potential games [Giannakopoulos, 2023, Giannakopoulos et al., 2022, Chen et al., 2020], as well as other important problems in game theory [Gatti et al., 2013, Buriol et al., 2011]. Other notable developments in a broader context were covered in an older survey by Spielman and Teng [2009]; for more recent developments, we point to, for example, Christ and Yannakakis [2023], Chen et al. [2024], Huiberts et al. [2023], and the many references therein.

Average-case analysis has also been a popular topic in the optimization literature [Cunha et al., 2022, Paquette et al., 2023, Scieur and Pedregosa, 2020], and so it is worth relating our results to that line of work. In particular, let us focus on the recent work of Cunha et al. [2022]. First, that paper targets a certain class of convex quadratic problems, whereas we examine zero-sum games. They also operate under a different perturbation model, deriving a parametrization based on the concentration of the eigenvalues of a certain matrix. Further, without strong convexity, Cunha et al. [2022] establish a complexity scaling with $\text{poly}(1/\epsilon)$, while here we target the $\log(1/\epsilon)$ regime. We finally remark that the techniques employed are also quite different. In particular, Cunha et al. [2022, Problem 2.1] posit that the optimal solution does not depend on the underlying randomization. In contrast, as we have already highlighted, the fact that the equilibrium is a function of the randomization constitutes the main technical crux in our setting. At the same time, Cunha et al. [2022] encountered several challenges not present in our setting, so overall those results are complementary.

Beyond smoothed complexity, understanding the last-iterate convergence of gradient-based methods such as OGDA and EGDA has received tremendous interest in the literature; *e.g.*, [Golowich et al., 2020a, Cai et al., 2022, Gorbunov et al., 2022, Vankov et al., 2023, Golowich et al., 2020b, Mahdavinia et al., 2022, Antonakopoulos et al., 2021, Mertikopoulos et al., 2019, Abe et al., 2023]. It is worth noting that linear convergence has also been documented for the more challenging class of extensive-form games [Lee et al., 2021], as well as Markov games [Song et al., 2023]. Nevertheless, there are lower bounds precluding linear convergence beyond affine variational inequalities [Golowich et al., 2020a, Wei et al., 2021]. We also refer to the works of Cohen et al. [2017] and Giannou et al. [2021] for further characterizations of the convergence rate of no-regret dynamics in multi-player games.

Contrary to the above line of work, which focuses on last-iterate convergence, the most common approach to solving zero-sum games revolves around regret minimization whereby optimality guarantees concern the average strategies. Learning in such settings has been a popular research topic as it captures many central problems; two notable recent applications are learning from multiple distributions [Haghtalab et al., 2022] and multi-calibration [Haghtalab et al., 2023]. Yet, there are at least three limitations of the no-regret framework worth highlighting here. The first one, which has been stressed extensively already, is that the number of iterations must grow at least as $\Omega(1/\epsilon)$ when one insists on taking (uniform) averages [Daskalakis et al., 2015]. The second and more nuanced caveat is that the no-regret framework does not provide instance-based guarantees based on natural game-theoretic parameters of the problem (see, for example, the discussion of Maiti

et al. [2023]). Building on earlier work [Wei et al., 2021, Tseng, 1995], some of our results here attempt to address this shortcoming by coming up with a more interpretable parameterization of the iteration complexity of algorithms such as `OGDA`. The final limitation is that, convergence to the set of equilibria notwithstanding, no-regret guarantees provide no information regarding properties of the equilibrium reached. Although not an issue in non-degenerate zero-sum games, equilibrium selection still remains a central problem. Earlier results [Wei et al., 2021, Tseng, 1995] provide an interesting characterization for the last iterate of `OGDA` and `EGDA` by showing that the limit point is the projection of the initial point to the set of equilibria.

Finally, it is worth pointing out the best available theoretical guarantees for solving zero-sum games. Assuming that each entry of $\mathbf{A}$ has absolute value bounded by 1, (1) can be solved in $\tilde{O}(\max\{n, m\}^\omega)$ [Cohen et al., 2021] or $\tilde{O}(nm + \min\{n, m\}^{5/2})$ [van den Brand et al., 2021]. Here, $\omega$ is the exponent of matrix multiplication and $\tilde{O}$ suppresses polylogarithmic factors in $n$ and $m$. The complexity we obtain for algorithms such as `OGDA` is not competitive even though we work in the more benign smoothed complexity model; we reiterate that we did not attempt to optimize the polynomial factors in terms of $n$ and $m$, and those can almost certainly be improved. On the other hand, there are two main aspects in which algorithms such as `OGDA` are more appealing in terms of their scalability: the per-iteration complexity and the memory requirements. An algorithm such as `OGDA` requires a single matrix-vector product in each iteration, which can be implemented in linear time for sparse matrices, and has a limited memory footprint. In contrast, implementing interior-point methods in large games can be prohibitive.

## B   Preliminaries

In this section, we introduce some further background on smoothed complexity and define the algorithms cited earlier (Items 1 to 4).

**Further notation**   For a random variable $X$, we denote by $\mathbb{E}[X]$ its expectation and by $\mathbb{V}[X]$ its variance, under the assumption that both are finite. For a sequence of random variables $X_1, \ldots, X_d$ and scalars $\alpha_1, \ldots, \alpha_d \in \mathbb{R}$, linearity of expectation yields that $\mathbb{E}[\alpha_1 X_1 + \cdots + \alpha_d X_d] = \alpha_1 \mathbb{E}[X_1] + \cdots + \alpha_d \mathbb{E}[X_d]$. Assuming independence, it also holds that $\mathbb{V}[\alpha_1 X_1 + \cdots + \alpha_d X_d] = (\alpha_1)^2 \mathbb{V}[X_1] + \cdots + (\alpha_d)^2 \mathbb{V}[X_d]$. We will also use the fact that a linear combination of independent Gaussian random variables is also Gaussian. More broadly, linear combinations can be understood through a convolution in the space of probability density functions, which means that smoothness (in the sense of Lemma C.7) is preserved in a certain regime.

### B.1   Smoothed complexity

To fully specify Definition 1.1, we first recall that a (univariate) Gaussian random variable with zero mean and variance $\sigma^2$ admits a probability density function of the form

$$\mu : t \mapsto \frac{1}{\sigma\sqrt{2\pi}} \exp\left(-\frac{t^2}{2\sigma^2}\right).$$

The law of such a Gaussian random variable will be denoted by $\mathcal{N}(0, \sigma^2)$. In the original work of Spielman and Teng [2004], smoothed complexity was defined as the expected running time (or some other cost function) of some algorithm over the perturbed input. More precisely, let $\mathcal{A}$ be an algorithm whose inputs can be expressed as vectors in $\mathbb{R}^d$, and let $T_\mathcal{A}(\mathcal{I})$ be the running time of algorithm $\mathcal{A}$ on input $\mathcal{I} \in \mathbb{R}^d$. Then, the *smoothed complexity* of $\mathcal{A}$ is

$$\mathcal{C}_\mathcal{A}(d, \sigma) := \max_{\mathcal{I} \in \mathbb{R}^d} \mathbb{E}_{\boldsymbol{g} \sim \mathcal{N}(\mathbf{0}_d, \sigma^2 \mathbf{I}_{d \times d})}[T_\mathcal{A}(\mathcal{I} + \|\mathcal{I}\| \boldsymbol{g})].$$

As pointed out by Spielman and Teng [2003], one does not need to limit smoothed analysis to measure the expected running time, and high probability guarantees are also quite natural; see, for example, the smoothed analysis of the perceptron algorithm due to Blum and Dunagan [2002]. Our main result also provides a guarantee with high probability; it is not clear whether the expected running time can also be bounded by $\mathrm{poly}(n, m, 1/\sigma)$, which is left for future work.

### B.2   Algorithms

Next, we specify the algorithms we consider in this work.

**Optimistic gradient descent/ascent**   Originally proposed by Popov [1980], optimistic gradient descent/ascent (OGDA)—and variants thereof [Hsieh et al., 2019]—has been recently revived in the online learning literature commencing from the pioneering works of Rakhlin and Sridharan [2013] and Chiang et al. [2012]. If we denote for compactness $F(\boldsymbol{z}) := (\mathbf{A}\boldsymbol{y}, -\mathbf{A}^\top \boldsymbol{x})$, OGDA can be expressed as follows for $t \in \mathbb{N}(= \{1, 2, \dots, \})$.

$$\begin{aligned} \boldsymbol{z}^{(t)} &:= \Pi_{\mathcal{Z}}(\widehat{\boldsymbol{z}}^{(t)} - \eta F(\boldsymbol{z}^{(t-1)})), \\ \widehat{\boldsymbol{z}}^{(t+1)} &:= \Pi_{\mathcal{Z}}(\widehat{\boldsymbol{z}}^{(t)} - \eta F(\boldsymbol{z}^{(t)})). \end{aligned} \tag{OGDA}$$

Here, $\eta > 0$ is the *learning rate*; $\Pi_{\mathcal{Z}}(\cdot)$ denotes the (Euclidean) projection operator on set $\mathcal{Z} := \mathcal{X} \times \mathcal{Y}$; and $\boldsymbol{z}^{(0)} = \widehat{\boldsymbol{z}}^{(1)} \in \mathcal{Z}$ is the initialization. That is, players simultaneously update their strategies through optimistic gradient steps. Given that $\mathcal{X}$ and $\mathcal{Y}$ are probability simplexes, each projection can be computed exactly in nearly linear time. The key reference point for OGDA in affine variational inequalities is the work of Wei et al. [2021] who established linear convergence using the notion of *metric subregularity* (Definition C.9), which is strongly related to Definition 1.3; we discuss their approach later in Appendix C.3.

**Optimistic multiplicative weights update**   Deriving from the same class of online learning algorithms as OGDA, optimistic multiplicative weights (OMWU) is the incarnation of *optimistic mirror descent* with an entropic regularizer, namely

$$\begin{aligned} \boldsymbol{x}^{(t)} &\propto \boldsymbol{x}^{(t-1)} \circ \exp\left(-2\eta \mathbf{A}\boldsymbol{y}^{(t-1)} + \eta \mathbf{A}\boldsymbol{y}^{(t-2)}\right), \\ \boldsymbol{y}^{(t)} &\propto \boldsymbol{y}^{(t-1)} \circ \exp\left(2\eta \mathbf{A}^\top \boldsymbol{x}^{(t-1)} - \eta \mathbf{A}^\top \boldsymbol{x}^{(t-2)}\right) \end{aligned} \tag{OMWU}$$

for $t \in \mathbb{N}$.[5] Above, $\circ$ denotes the component-wise product; the exponential mapping $\exp(\cdot)$ is also to be applied component-wise; and $\boldsymbol{z}^{(-1)} := \boldsymbol{z}^{(0)} := (\frac{1}{n}\mathbf{1}_n, \frac{1}{m}\mathbf{1}_m)$. Daskalakis and Panageas [2019] first proved that OMWU exhibits asymptotic (last-iterate) convergence, and Wei et al. [2021] later established linear convergence.

*Remark* B.1. It is important to note here that the exponential map of OMWU can produce iterates with an arbitrarily large number of bits. Nevertheless, it is not hard to show that the analysis of Wei et al. [2021] carries over when the iterates are truncated up to a certain length of the most significant bits, and so we will not dwell further on this issue here.

**Extra-gradient descent/ascent**   The extra-gradient method of Korpelevich [1976] is quite similar to OGDA, namely

$$\begin{aligned} \widehat{\boldsymbol{z}}^{(t)} &:= \Pi_{\mathcal{Z}}(\boldsymbol{z}^{(t)} - \eta F(\boldsymbol{z}^{(t)})), \\ \boldsymbol{z}^{(t+1)} &:= \Pi_{\mathcal{Z}}(\boldsymbol{z}^{(t)} - \eta F(\widehat{\boldsymbol{z}}^{(t)})) \end{aligned} \tag{EGDA}$$

for $t \in \mathbb{N}$. Unlike OGDA, one caveat is that it requires two gradient evaluations per each iteration $t$. EGDA is also less suited to use in an online environment: it requires more feedback than what is provided in the online learning setting, and in fact, even legitimate variants of EGDA can still incur substantial regret [Golowich et al., 2020a]. Tseng [1995] first established that EGDA exhibits linear convergence for problems such as (1), discussed further in Appendix C.3.

**Iterative smoothing**   This is a refinement of Nesterov's classical smoothing technique [Nesterov, 2005] due to Gilpin et al. [2012]. Let us first recall the vanilla version of Nesterov, which we refer to as $\texttt{Smoothing}(\mathbf{A}, \boldsymbol{z}^{(0)}, \epsilon)$:

1. Initialize $\eta := \frac{\epsilon}{D_{\mathcal{Z}}}$ and $\widehat{\boldsymbol{z}}^{(0)} := \boldsymbol{z}^{(0)}$, where $D_{\mathcal{Z}}$ is the $\ell_2$ diameter of $\mathcal{Z}$.
2. For $t = 0, 1, \dots$
    (a) $\boldsymbol{u}^{(t)} := \frac{2}{2+t}\widehat{\boldsymbol{z}}^{(t)} + \frac{t}{t+2}\boldsymbol{z}^{(t)}$.
    (b)
    $$\boldsymbol{z}^{(t+1)} := \arg\min_{\boldsymbol{z} \in \mathcal{Z}} \left\{ \langle \nabla F_\eta(\boldsymbol{u}^{(t)}), \boldsymbol{z} - \boldsymbol{u}^{(t)} \rangle + \frac{L^2}{2\eta}\|\boldsymbol{z} - \boldsymbol{u}^{(t)}\|^2 \right\},$$

---

[5]OMWU is oftentimes expressed via the (optimistic) mirror descent viewpoint, but the form we provide here is easily seen to be equivalent.

where $F_\eta(z) := \max_{\widehat{z} \in \mathcal{Z}} \{ \langle F(z), z - \widehat{z} \rangle - \frac{\eta}{2} \| z - \widehat{z} \|^2 \}$ and $L$ is a suitable matrix norm.

(c) If $\Phi(z^{(t+1)}) < \epsilon$, **return**.

(d)
$$\widehat{z}^{(t+1)} := \arg\min_{\widehat{z} \in \mathcal{Z}} \left\{ \sum_{\tau=0}^{t} \frac{\tau+1}{2} \langle \nabla F_\eta(u^{(\tau)}), \widehat{z} - u^{(\tau)} \rangle + \frac{L^2}{2\eta} \| \widehat{z} - z^{(0)} \|^2 \right\}.$$

In this context, $\texttt{IterSmooth}(\mathbf{A}, z^{(0)}, \rho, \epsilon)$ is simple refinement of $\texttt{Smoothing}$, which nonetheless attains linear convergence [Gilpin et al., 2012].

1. Let $\epsilon^{(0)} = F(z^{(0)})$.
2. For $t = 0, 1, \ldots$
   (a) $\epsilon^{(t+1)} := \frac{\epsilon^{(t)}}{\rho}$.
   (b) $z^{(t+1)} := \texttt{Smoothing}(\mathbf{A}, z^{(t)}, \epsilon^{(t+1)})$.
   (c) If $\Phi(z^{(t+1)}) < \epsilon$, **return**.

## C  Omitted proofs

We dedicate this section to the proofs omitted earlier from the main body.

### C.1  Proofs from Section 3.2

We first point out that degenerates games have measure zero (*cf.* Spielman and Teng [2003, Proposition 5.1]).

**Lemma C.1.** *For a Gaussian distributed payoff matrix* $\mathbf{A}$ *per Definition 1.1, the game is non-degenerate (Definition 3.2) with probability* 1 *(almost surely).*

Indeed, the set of games with a non unique equilibrium has measure zero [van Damme, 1991, Theorem 3.5.1]. Regarding the characterization in terms of the number of tight inequalities of the corresponding (primal and dual) linear programs, gathered in Definition 3.2, we note that if $n + 1$ of the inequalities were tight at $x^\star$, that would induce a feasible linear system of $n$ equalities (by eliminating $v$) in $n - 1$ variables (by eliminating one of the redundant variables); such degeneracies have measure zero, and there are only finitely many possible such degeneracies, leading to Lemma C.1. As a result, in the smoothed complexity model, we can safely assume that the game is non-degenerate.

Now, as we alluded to earlier, establishing Definition 1.3 reduces to showing that for any points $x \in \mathcal{X}$ and $y \in \mathcal{Y}$,

$$\max_{y' \in \mathcal{Y}} \langle x, \mathbf{A}y' \rangle - v \geq \kappa \| x - \Pi_{\mathcal{X}^\star}(x) \| = \kappa \| x - x^\star \|, \tag{8}$$

$$v - \min_{x' \in \mathcal{X}} \langle x', \mathbf{A}y \rangle \geq \kappa \| y - \Pi_{\mathcal{Y}^\star}(y) \| = \kappa \| y - y^\star \|. \tag{9}$$

(Definition 1.3 then indeed follows from the obvious fact $\| x - x^\star \| + \| y - y^\star \| \geq \| z - z^\star \|$.) Accordingly, our proof of Theorem 3.6 below will focus on lower bounding $\kappa$ so that (8) holds, and (9) can then be treated similarly.

Before we proceed, let us make some observations regarding transformation (5) we saw earlier. First, one can understand the transformation $\mathbf{A}_{B,N}^\flat = \mathbf{T}(\mathbf{Q}^\flat, b, c, d)$ through the equations

$$d = \mathbf{A}_{i,j}; \, b_{j'} = -\mathbf{A}_{i,j'} + \mathbf{A}_{i,j}; \, c_{i'} = -\mathbf{A}_{i',j} + \mathbf{A}_{i,j}; \, \mathbf{Q}_{i',j'} = \mathbf{A}_{i',j'} - \mathbf{A}_{i,j'} - \mathbf{A}_{i',j} + \mathbf{A}_{i,j} \tag{10}$$

for all $(i', j') \in \widetilde{B} \times \widetilde{N}$. This can easily be derived from (5) by using the fact that $\widehat{x}_B = (\widetilde{x}, 1 - \mathbf{1}^\top \widetilde{x})$ and $\widehat{y}_N = (\widetilde{y}, 1 - \mathbf{1}^\top \widetilde{y})$. From (10), we see that there is a permutation of the rows of $\mathbf{T}$ that is upper triangular, with every entry being either 1 or $-1$. This implies that $|\det(\mathbf{T})| = 1$. With a slight abuse of notation, we will write $\mathbf{T}_{i,j}$ (as opposed to $\mathbf{T}_{(i,j),:}$) to access the $(i, j)$ row of $\mathbf{T}$, so that $\mathbf{A}_{i,j} = \langle \mathbf{T}_{i,j}, (\mathbf{Q}^\flat, b, c, d) \rangle$. From (10), we also see that $\mathbf{T}_{i,j}$ contains at most 4 non-zero entries. In turn, this implies that $\| \mathbf{T}_{i,j} \| \leq 2$ and $\| \mathbf{T}_{i,j} \|_1 \leq 4$. We gather the above observations in the claim below, which will be used in the sequel.

**Claim C.2.** *For the (linear) transformation* $\mathbf{T} \in \mathbb{R}^{(BN)\times(BN)}$ *given in* (10), *it holds that* $|\det(\mathbf{T})| = 1$. *Further,* $\|\mathbf{T}_{i,j}\| \leq 2$ *and* $\|\mathbf{T}_{i,j}\|_1 \leq 4$ *for all* $(i,j) \in B \times N$.

The point of transformation (5) is that, as we claimed earlier, the spectral properties of matrix $\mathbf{Q}$ (as opposed to $\mathbf{A}_{B,N}$, which is a natural candidate) suffice to capture the difficulty of addressing the second subproblem identified in Section 3.2. In addition, there is a straightforward but convenient characterization of the equilibrium $(\boldsymbol{x}^\star, \boldsymbol{y}^\star)$ in terms of the transformed game in (5), as stated below.

**Claim C.3.** *It holds that* $\mathbf{Q}\widetilde{\boldsymbol{y}}^\star = \boldsymbol{c}$ *and* $\mathbf{Q}^\top \widetilde{\boldsymbol{x}}^\star = \boldsymbol{b}$.

*Proof.* It is clear that the vector $\mathbf{Q}\widetilde{\boldsymbol{y}}^\star - \boldsymbol{c}$ must have the same value in every coordinate since $\widetilde{\boldsymbol{x}}^\star$ is fully supported and a best response (by assumption). If that entry was positive, then $\widetilde{\boldsymbol{x}}^\star$ would not be a best response since Player $x$ could profit from removing all the probability mass (which is possible since $\sum_{i\in\widetilde{B}} \boldsymbol{x}_i^\star > 0$). If there was a negative entry, Player $x$ would profit from increasing its probability mass (which is possible since $\sum_{i\in\widetilde{B}} \boldsymbol{x}_i^\star < 1$). Similar reasoning yields $\mathbf{Q}^\top \widetilde{\boldsymbol{x}}^\star = \boldsymbol{b}$. $\square$

Having made the above observations, we next prove some lemmas claimed earlier in Section 3.2 which will be used for the proof of Theorem 3.6. First, we give the proof of Lemma 3.4.

**Lemma 3.4.** *Let* $\boldsymbol{c} = \mathbf{Q}\widetilde{\boldsymbol{y}}^\star = \sum_{j\in\widetilde{N}} \widetilde{\boldsymbol{y}}_j^\star \mathbf{Q}_{:,j}$, *and suppose that* $\overline{\mathbf{Q}} \in \mathbb{R}^{\widetilde{B}\times\widetilde{N}}$ *is such that its $j$th column is equal to* $\mathbf{Q}_{:,j} - \boldsymbol{c}$. *Then,*

$$
\min_{j\in\widetilde{N}} \mathsf{dist}(\mathbf{Q}_{:,j}, \mathsf{span}(\mathbf{Q}_{:,\widetilde{N}-j})) \leq \left(1 + \frac{|\widetilde{N}|}{1 - \sum_{j\in\widetilde{N}} \boldsymbol{y}_j^\star}\right) \min_{j\in\widetilde{N}} \mathsf{dist}(\overline{\mathbf{Q}}_{:,j}, \mathsf{span}(\overline{\mathbf{Q}}_{:,\widetilde{N}-j})).
$$

*Proof.* Let $\widetilde{N} \ni j' \in \arg\min_{j\in\widetilde{N}} \mathsf{dist}(\overline{\mathbf{Q}}_{:,j}, \mathsf{span}(\overline{\mathbf{Q}}_{:,\widetilde{N}-j}))$. By definition, there is $\boldsymbol{\rho} \in \mathbb{R}^{\widetilde{N}-j'}$ and $\boldsymbol{r} \in \mathbb{R}^{\widetilde{N}}$ with $\|\boldsymbol{r}\| = 1$ such that

$$
\overline{\mathbf{Q}}_{:,j} := -\sum_{j\in\widetilde{N}-j'} \widetilde{\boldsymbol{y}}_j^\star \mathbf{Q}_{:,j} + (1 - \boldsymbol{y}_{j'}^\star)\mathbf{Q}_{:,j'} = \sum_{j\in\widetilde{N}-j'} \boldsymbol{\rho}_j(\mathbf{Q}_{:,j} - \boldsymbol{c}) + \epsilon\boldsymbol{r},
$$

where $\epsilon := \min_{j\in\widetilde{N}} \mathsf{dist}(\overline{\mathbf{Q}}_{:,j}, \mathsf{span}(\overline{\mathbf{Q}}_{:,\widetilde{N}-j}))$. Rearranging, we have

$$
\overbrace{\mathbf{Q}_{:,j'}\left(1 - \boldsymbol{y}_{j'}^\star + \boldsymbol{y}_{j'}^\star \sum_{j\in\widetilde{N}-j'} \boldsymbol{\rho}_j\right)}^{\phi_{j'}} + \sum_{j\in\widetilde{N}-j'}\overbrace{\mathbf{Q}_{:,j}\left(-\boldsymbol{y}_j^\star - \boldsymbol{\rho}_j + \boldsymbol{y}_j^\star \sum_{j''\in\widetilde{N}-j'}\boldsymbol{\rho}_{j''}\right)}^{\phi_j} = \epsilon\boldsymbol{r}. \quad (11)
$$

Now, let us suppose that all coefficients above are such that $|\phi_j| \leq \epsilon' := \frac{1 - \sum_{j\in\widetilde{N}} \boldsymbol{y}_j^\star}{1 - \sum_{j\in\widetilde{N}} \boldsymbol{y}_j^\star + |\widetilde{N}|}$ for all $j \in \widetilde{N}$. Then, $\sum_{j\in\widetilde{N}} \phi_j = \pm|\widetilde{N}|\epsilon'$ since $|\sum_{j\in\widetilde{N}} \phi_j| \leq \sum_{j\in\widetilde{N}} |\phi_j| \leq \epsilon|\widetilde{N}|$, where for convenience we used the notation $\sum_{j\in\widetilde{N}} \phi_j = \pm|\widetilde{N}|\epsilon' \iff -|\widetilde{N}|\epsilon' \leq \sum_{j\in\widetilde{N}} \phi_j \leq |\widetilde{N}|\epsilon'$. Thus, by definition of $\phi_j$,

$$
\left(1 - \sum_{j\in\widetilde{N}} \boldsymbol{y}_j^\star\right)\left(\sum_{j\in\widetilde{N}-j'} \boldsymbol{\rho}_j\right) = \left(1 - \sum_{j\in\widetilde{N}} \boldsymbol{y}_j^\star\right) \pm \epsilon'|\widetilde{N}|.
$$

Since $0 < 1 - \sum_{j\in\widetilde{N}} \boldsymbol{y}_j^\star$, we have

$$
\left(\sum_{j\in\widetilde{N}-j'} \boldsymbol{\rho}_j\right) = 1 \pm \epsilon' \frac{|\widetilde{N}|}{1 - \sum_{j\in\widetilde{N}} \boldsymbol{y}_j^\star}.
$$

Thus,

$$
\phi_{j'} = 1 - \boldsymbol{y}_{j'}^\star + \boldsymbol{y}_{j'}^\star \sum_{j\in\widetilde{N}-j'} \boldsymbol{\rho}_j = 1 \pm \epsilon' \frac{|\widetilde{N}|}{1 - \sum_{j\in\widetilde{N}} \boldsymbol{y}_j^\star} > \epsilon'
$$

since $\epsilon' \leq \frac{1-\sum_{j\in\widetilde{N}}\boldsymbol{y}_j^\star}{1-\sum_{j\in\widetilde{N}}\boldsymbol{y}_j^\star+|\widetilde{N}|}$. The last displayed inequality contradicts our earlier assumption that $|\phi_{j'}| \leq \epsilon'$. As a result, we conclude that at least one coefficient $\phi_j$ has an absolute value at least $\epsilon'$. Dividing (11) by that coefficient, we get

$$\min_{j\in\widetilde{N}}\mathsf{dist}(\mathbf{Q}_{:,j},\mathsf{span}(\mathbf{Q}_{:,\widetilde{N}-j})) \leq \frac{\epsilon}{\epsilon'} \leq \left(1 + \frac{|\widetilde{N}|}{1-\sum_{j\in\widetilde{N}}\boldsymbol{y}_j^\star}\right)\min_{j\in\widetilde{N}}\mathsf{dist}(\overline{\mathbf{Q}}_{:,j},\mathsf{span}(\overline{\mathbf{Q}}_{:,\widetilde{N}-j})).$$

This completes the proof. $\qquad\square$

We continue with the proof of Lemma 3.5.

**Lemma 3.5.** *Let $\mathbf{M} \in \mathbb{R}^{d\times d}$ be a full-rank matrix. For any $\widetilde{\boldsymbol{x}} \in \mathbb{R}^d$ there is $\boldsymbol{p} \in \mathbb{R}^d$ with $\|\boldsymbol{p}\| \leq \frac{1}{\sigma_{\min}(\mathbf{M})}\|\widetilde{\boldsymbol{x}}\|$ such that*

$$\widetilde{\boldsymbol{x}} = \mathbf{M}\boldsymbol{p} = \sum_{j=1}^{d}\boldsymbol{p}_j\mathbf{M}_{:,j}.$$

*Proof.* Let $\mathbf{M} = \mathbf{U}\boldsymbol{\Sigma}\mathbf{V}^\top$ be a singular value decomposition (SVD) of $\mathbf{Q}$, where $\mathbf{U}$ and $\mathbf{V}$ are orthonormal. Then, given that $\mathbf{Q}$ is invertible (by assumption),

$$\boldsymbol{p} = \mathbf{V}\boldsymbol{\Sigma}^{-1}\mathbf{U}^\top\widetilde{\boldsymbol{x}},$$

where $\boldsymbol{\Sigma}^{-1} = \mathsf{diag}(\sigma_{\min}^{-1},\ldots,\sigma_{\max}^{-1})$. (Here, $\sigma_{\max}$ and $\sigma_{\min}$ are the maximum and minimum singular values of $\mathbf{M}$, respectively.) Thus, $\|\boldsymbol{p}\| \leq \|\mathbf{V}\|\|\boldsymbol{\Sigma}^{-1}\|\|\mathbf{U}^\top\|\|\widetilde{\boldsymbol{x}}\| \leq \frac{1}{\sigma_{\min}(\mathbf{Q})}\|\widetilde{\boldsymbol{x}}\|$, where we used the fact that the spectral norm of any orthonormal matrix is $1$ and the spectral norm of any diagonal matrix is its maximum entry in absolute value. $\qquad\square$

We next state the negative second moment identity that connects the smallest singular values in terms of a certain geometric property of the matrix (namely, Item 3) (see also [Tao, 2023] for further background).

**Proposition C.4** (Negative second moment identity [Tao et al., 2010])**.** *Let $\mathbf{M} \in \mathbb{R}^{d\times d}$ be an invertible matrix. Then,*

$$\sum_{r=1}^{d}\frac{1}{\sigma_r^2(\mathbf{M})} = \sum_{r=1}^{d}\frac{1}{\mathsf{dist}^2(\mathbf{M}_{r,:},H_{-r,:})} = \sum_{r=1}^{d}\frac{1}{\mathsf{dist}^2(\mathbf{M}_{:,r},H_{:,-r})}, \tag{12}$$

*where $H_{-r,:} := \mathsf{span}(\mathbf{M}_{1,:},\ldots,\mathbf{M}_{r-1,:},\mathbf{M}_{r+1,:},\ldots,\mathbf{M}_{d,:})$.*

One can readily prove this identity by equivalently expressing the negative second moment $\mathsf{tr}((\mathbf{M}^{-1})^\top\mathbf{M}^{-1})$ as either $\sum_{r=1}^{d}\sigma_r^2(\mathbf{M}^{-1}) = \sum_{r=1}^{d}\sigma_r^{-2}(\mathbf{M})$ or $\sum_{r=1}^{d}\|\mathbf{M}_{:,r}^{-1}\|^2$, leading to the first identity in (12). The second one follows from the fact that the singular values of $\mathbf{M}^\top$ coincide with the singular values of $\mathbf{M}$.

We are now ready to prove Theorem 3.6, restated below.

**Theorem 3.6.** *Let $\mathbf{A}$ be a non-degenerate payoff matrix, and suppose that $(\alpha_P(\mathbf{A}),\alpha_D(\mathbf{A}))$, $(\beta_P(\mathbf{A}),\beta_D(\mathbf{A}))$ and $(\gamma_P(\mathbf{A}),\gamma_D(\mathbf{A}))$ are as in Definition 3.3. Then, the error bound (Definition 1.3) is satisfied for any sufficiently small modulus*

$$\kappa \gtrsim \frac{1}{\|\mathbf{A}^\flat\|_\infty}\frac{1}{\min(n,m)^3}\min\left\{(\alpha_D(\mathbf{A}))^2\beta_D(\mathbf{A})\gamma_P(\mathbf{A}),(\alpha_P(\mathbf{A}))^2\beta_P(\mathbf{A})\gamma_D(\mathbf{A})\right\}.$$

*Proof.* We lower bound $\kappa$ so that (8) holds; bound (9) will then be treated in a symmetric fashion, and Definition 1.3 will follow.

Let us fix any point $\boldsymbol{x} \in \mathcal{X}$. We can write $\boldsymbol{x}$ as $\lambda\widehat{\boldsymbol{x}}_B + (1-\lambda)\widehat{\boldsymbol{x}}_{\overline{B}}$ for some $\lambda \in [0,1]$ such that $\widehat{\boldsymbol{x}}_B \in \mathcal{X}$ and all coordinates of $\widehat{\boldsymbol{x}}_B$ in $\overline{B}$ are zero, and $\widehat{\boldsymbol{x}}_{\overline{B}} \in \mathcal{X}$ and all coordinates of $\widehat{\boldsymbol{x}}_{\overline{B}}$ in $B$ are zero. For notational convenience, we define

$$P(\mathbf{A}) := \frac{1}{2|N|\sqrt{|B|}}\sigma_{\min}(\overline{\mathbf{Q}})\left(1 + \frac{1}{\alpha_D(\mathbf{A})}\right)^{-1}. \tag{13}$$

We consider the following two cases.

**Case I:** $\lambda P(\mathbf{A}) \| \widehat{\boldsymbol{x}}_B - \boldsymbol{x}_B^\star \| \geq 4(1 - \lambda) \| \mathbf{A}^\flat \|_\infty$. If $\widehat{\boldsymbol{x}}_B = \boldsymbol{x}_B^\star$, it follows that $\boldsymbol{x} = \boldsymbol{x}^\star$ (since $\lambda = 1$), and the conclusion trivially follows. We can thus assume that $\widehat{\boldsymbol{x}}_B \neq \boldsymbol{x}_B^\star$. In this case, it follows that $\widetilde{B} \neq \emptyset$, and we proceed as follows.

$$\max_{\boldsymbol{y}' \in \mathcal{Y}} \langle \boldsymbol{x}, \mathbf{A}\boldsymbol{y}' \rangle - v \geq \lambda \max_{j \in N} \langle \widehat{\boldsymbol{x}}_B - \boldsymbol{x}_B^\star, \mathbf{A}_{B,j} \rangle + (1 - \lambda) \left( \langle \boldsymbol{x}_{\overline{B}}, \mathbf{A}_{\overline{B},j} \rangle - v \right) \tag{14}$$

$$\geq \lambda \max_{j \in N} \langle \widehat{\boldsymbol{x}}_B - \boldsymbol{x}_B^\star, \mathbf{A}_{B,j} \rangle - 2(1 - \lambda) \| \mathbf{A}^\flat \|_\infty, \tag{15}$$

where (14) follows from the definition $\boldsymbol{x} := \lambda \widehat{\boldsymbol{x}}_B + (1 - \lambda) \widehat{\boldsymbol{x}}_{\overline{B}}$ and the fact that $v = \langle \boldsymbol{x}_B^\star, \mathbf{A}_{B,j} \rangle$ for all $j \in N$; and (15) uses definition of $\| \mathbf{A}^\flat \|_\infty$ to lower bound the second term in (14). Continuing from (15), we can use the transformation defined in (5) to get

$$\max_{j \in N} \langle \widehat{\boldsymbol{x}}_B - \boldsymbol{x}_B^\star, \mathbf{A}_{B,j} \rangle = \max_{j \in N} \langle \widetilde{\boldsymbol{x}} - \widetilde{\boldsymbol{x}}^\star, \mathbf{Q}_{:,j} - \boldsymbol{c} \rangle, \tag{16}$$

where, with an abuse of notation, the convention above is that $\mathbf{Q}_{:,j} = \mathbf{0}$ if $j \neq \widetilde{N}$. For convenience, let us define $\chi_j := \langle \widetilde{\boldsymbol{x}} - \widetilde{\boldsymbol{x}}^\star, \mathbf{Q}_{:,j} - \boldsymbol{c} \rangle$ for all $j \in N$. Our goal is to lower bound $\max_{j \in N} \chi_j$. To that end, we first observe that, by the fact that $\mathbf{Q}\widetilde{\boldsymbol{y}}^\star = \boldsymbol{c}$ (Claim C.3),

$$0 = \langle \widetilde{\boldsymbol{x}} - \widetilde{\boldsymbol{x}}^\star, \mathbf{Q}\widetilde{\boldsymbol{y}}^\star - \boldsymbol{c} \rangle = \sum_{j \in \widetilde{N}} \widetilde{\boldsymbol{y}}_j^\star \langle \widetilde{\boldsymbol{x}} - \widetilde{\boldsymbol{x}}^\star, \mathbf{Q}_{:,j} \rangle - \langle \widetilde{\boldsymbol{x}} - \widetilde{\boldsymbol{x}}^\star, \boldsymbol{c} \rangle$$

$$= \sum_{j \in \widetilde{N}} \widetilde{\boldsymbol{y}}_j^\star \langle \widetilde{\boldsymbol{x}} - \widetilde{\boldsymbol{x}}^\star, \mathbf{Q}_{:,j} - \boldsymbol{c} \rangle + \left( 1 - \sum_{j \in \widetilde{N}} \widetilde{\boldsymbol{y}}_j^\star \right) \langle \widetilde{\boldsymbol{x}} - \widetilde{\boldsymbol{x}}^\star, -\boldsymbol{c} \rangle.$$

In other words,

$$\sum_{j \in N} \boldsymbol{y}_j^\star \chi_j = 0,$$

which in turn implies that

$$\sum_{j \in N} \max(0, \chi_j) \geq \sum_{j \in N} \boldsymbol{y}_j^\star \max(0, \chi_j) = -\sum_{j \in N} \boldsymbol{y}_j^\star \min(0, \chi_j)$$

$$\geq -\alpha_D(\mathbf{A}) \sum_{j \in N} \min(0, \chi_j), \tag{17}$$

where we made use of the obvious identity $t = \max(0, t) + \min(0, t)$ for all $t \in \mathbb{R}$, as well as the definition of $\alpha_D(\mathbf{A})$ (Item 1). We let $\boldsymbol{p} \in \mathbb{R}^{\widetilde{N}}$ be the (unique) solution to the linear system

$$\widetilde{\boldsymbol{x}} - \widetilde{\boldsymbol{x}}^\star = \overline{\mathbf{Q}}\boldsymbol{p} = \sum_{j \in \widetilde{N}} (\mathbf{Q}_{:,j} - \boldsymbol{c})\boldsymbol{p}_j,$$

and $\boldsymbol{p}_j = 0$ for $j \in N \setminus \widetilde{N}$. By Lemma 3.5, we know that $\| \boldsymbol{p} \| \leq (\sigma_{\min}(\overline{\mathbf{Q}}))^{-1} \| \widetilde{\boldsymbol{x}} - \widetilde{\boldsymbol{x}}^\star \|$. Then, we have

$$\sum_{j \in N} \chi_j \boldsymbol{p}_j = \sum_{j \in \widetilde{N}} \chi_j \boldsymbol{p}_j = \left\langle \widetilde{\boldsymbol{x}} - \widetilde{\boldsymbol{x}}^\star, \sum_{j \in \widetilde{N}} (\mathbf{Q}_{:,j} - \boldsymbol{c})\boldsymbol{p}_j \right\rangle = \| \widetilde{\boldsymbol{x}} - \widetilde{\boldsymbol{x}}^\star \|^2. \tag{18}$$

Moreover,

$$\sum_{j \in N} \chi_j \boldsymbol{p}_j = \sum_{j \in N} \boldsymbol{p}_j \max(0, \chi_j) + \sum_{j \in N} \boldsymbol{p}_j \min(0, \chi_j)$$

$$\leq \sum_{j \in N} \max(0, \boldsymbol{p}_j) \max(0, \chi_j) + \sum_{j \in N} \min(0, \boldsymbol{p}_j) \min(0, \chi_j) \tag{19}$$

$$\leq \| \boldsymbol{p} \|_\infty \sum_{j \in N} \max(0, \chi_j) - \| \boldsymbol{p} \|_\infty \sum_{j \in N} \min(0, \chi_j) \tag{20}$$

$$\leq \| \boldsymbol{p} \|_\infty \left( 1 + \frac{1}{\alpha_D(\mathbf{A})} \right) \sum_{j \in N} \max(0, \chi_j) \tag{21}$$

$$\leq \frac{1}{\sigma_{\min}(\overline{\mathbf{Q}})} \left( 1 + \frac{1}{\alpha_D(\mathbf{A})} \right) |N| \max_{j \in N} \chi_j \| \widetilde{\boldsymbol{x}} - \widetilde{\boldsymbol{x}}^\star \|, \tag{22}$$

where (19) follows from the fact that $\boldsymbol{p}_j \max(0, \chi_j) \leq \max(0, \boldsymbol{p}_j) \max(0, \chi_j)$ (by nonnegativity of $\max(0, \chi_j)$) and $\boldsymbol{p}_j \min(0, \chi_j) \leq \min(0, \boldsymbol{p}_j) \min(0, \chi_j)$ (by nonpositivity of $\min(0, \chi_j)$); (20) uses that $\min(0, \boldsymbol{p}_j) \geq -|\boldsymbol{p}_j| \geq -\|\boldsymbol{p}\|_\infty$, which gives $\min(0, \boldsymbol{p}_j) \min(0, \chi_j) \leq -\|\boldsymbol{p}\|_\infty \min(0, \chi_j)$; (21) follows from (17); and (22) uses the bound $\|\boldsymbol{p}\|_2 \leq (\sigma_{\min}(\overline{\mathbf{Q}}))^{-1}\|\widetilde{\boldsymbol{x}} - \widetilde{\boldsymbol{x}}^\star\|$ (Lemma 3.5). Combining (18) and (22),

$$\max_{j \in N}\langle \widehat{\boldsymbol{x}}_B - \boldsymbol{x}_B^\star, \mathbf{A}_{B,j}\rangle \geq \frac{1}{|N|}\sigma_{\min}(\overline{\mathbf{Q}})\left(1 + \frac{1}{\alpha_D(\mathbf{A})}\right)^{-1}\|\widetilde{\boldsymbol{x}} - \widetilde{\boldsymbol{x}}^\star\| \tag{23}$$

$$\geq \frac{1}{2|N|\sqrt{|B|}}\sigma_{\min}(\overline{\mathbf{Q}})\left(1 + \frac{1}{\alpha_D(\mathbf{A})}\right)^{-1}\|\widehat{\boldsymbol{x}}_B - \boldsymbol{x}_B^\star\|, \tag{24}$$

where (23) uses the definition of $\chi_j$ and the assumption that $\widetilde{\boldsymbol{x}} \neq \widetilde{\boldsymbol{x}}^\star$ (equivalently, $\boldsymbol{x}_B^\star \neq \widehat{\boldsymbol{x}}_B$), and (24) follows from the bound

$$\|\widehat{\boldsymbol{x}}_B - \boldsymbol{x}_B^\star\| \leq \|\widehat{\boldsymbol{x}}_B - \boldsymbol{x}_B^\star\|_1 \leq \sum_{i \in \widetilde{B}}|\boldsymbol{x}_i - \boldsymbol{x}_i^\star| + \left|\sum_{i \in \widetilde{B}}(\boldsymbol{x}_i - \boldsymbol{x}_i^\star)\right| \leq 2\|\widetilde{\boldsymbol{x}} - \widetilde{\boldsymbol{x}}^\star\|_1 \leq 2\sqrt{|B|}\|\widetilde{\boldsymbol{x}} - \widetilde{\boldsymbol{x}}^\star\|.$$

Returning to (15), we have

$$\max_{\boldsymbol{y}' \in \mathcal{Y}}\langle \boldsymbol{x}, \mathbf{A}\boldsymbol{y}'\rangle - v \geq \lambda\frac{1}{2|N|\sqrt{|B|}}\sigma_{\min}(\overline{\mathbf{Q}})\left(1 + \frac{1}{\alpha_D(\mathbf{A})}\right)^{-1}\|\widehat{\boldsymbol{x}}_B - \boldsymbol{x}_B^\star\| - 2(1 - \lambda)\|\mathbf{A}^\flat\|_\infty$$

$$= \lambda P(\mathbf{A})\|\widehat{\boldsymbol{x}}_B - \boldsymbol{x}_B^\star\| - 2(1 - \lambda)\|\mathbf{A}^\flat\|_\infty, \tag{25}$$

where the equality above follows from the definition of $P(\mathbf{A})$ in (13). Next, we bound

$$\|\boldsymbol{x} - \boldsymbol{x}^\star\|^2 = \|\lambda\widehat{\boldsymbol{x}}_B - \boldsymbol{x}_B^\star\|^2 + (1 - \lambda)^2\|\widehat{\boldsymbol{x}}_{\overline{B}}\|^2$$

$$= \|\lambda(\widehat{\boldsymbol{x}}_B - \boldsymbol{x}_B^\star) - (1 - \lambda)\boldsymbol{x}_B^\star\|^2 + (1 - \lambda)^2\|\widehat{\boldsymbol{x}}_{\overline{B}}\|^2$$

$$\leq 2\lambda^2\|\widehat{\boldsymbol{x}}_B - \boldsymbol{x}_B^\star\|^2 + 2(1 - \lambda)^2\|\boldsymbol{x}_B^\star\|^2 + (1 - \lambda)^2\|\widehat{\boldsymbol{x}}_{\overline{B}}\|^2 \tag{26}$$

$$\leq 2\lambda^2\|\widehat{\boldsymbol{x}}_B - \boldsymbol{x}_B^\star\|^2 + 3(1 - \lambda)^2, \tag{27}$$

where (26) uses triangle inequality with respect to $\|\cdot\|$ along with the inequality $(t_1 + t_2)^2 \leq 2t_1^2 + 2t_2^2$, and (27) uses that $\|\boldsymbol{x}_B^\star\|, \|\widehat{\boldsymbol{x}}_{\overline{B}}\| \leq 1$. Since we are assuming that $\lambda P(\mathbf{A})\|\widehat{\boldsymbol{x}}_B - \boldsymbol{x}_B^\star\| \geq 4(1 - \lambda)\|\mathbf{A}^\flat\|_\infty$, (27) in turn implies that

$$\|\boldsymbol{x} - \boldsymbol{x}^\star\|^2 \leq 2\lambda^2\|\widehat{\boldsymbol{x}}_B - \boldsymbol{x}_B^\star\|^2 + \lambda^2\left(\frac{P(\mathbf{A})}{\|\mathbf{A}^\flat\|_\infty}\right)^2\|\widehat{\boldsymbol{x}}_B - \boldsymbol{x}_B^\star\|^2$$

$$= \lambda^2\left(2 + \left(\frac{P(\mathbf{A})}{\|\mathbf{A}^\flat\|_\infty}\right)^2\right)\|\widehat{\boldsymbol{x}}_B - \boldsymbol{x}_B^\star\|^2. \tag{28}$$

Combining (25) and (28) with the assumption that $\lambda P(\mathbf{A})\|\widehat{\boldsymbol{x}}_B - \boldsymbol{x}_B^\star\| \geq 4(1 - \lambda)\|\mathbf{A}^\flat\|_\infty$,

$$\max_{\boldsymbol{y}' \in \mathcal{Y}}\langle \boldsymbol{x}, \mathbf{A}\boldsymbol{y}'\rangle - v \geq \frac{\lambda}{2}P(\mathbf{A})\|\widehat{\boldsymbol{x}}_B - \boldsymbol{x}_B^\star\|$$

$$\geq \frac{1}{2}P(\mathbf{A})\left(2 + \left(\frac{P(\mathbf{A})}{\|\mathbf{A}^\flat\|_\infty}\right)^2\right)^{-2}\|\boldsymbol{x} - \boldsymbol{x}^\star\| \geq \kappa(\mathbf{A})\|\boldsymbol{x} - \boldsymbol{x}^\star\|.$$

It is easy to see that $P(\mathbf{A})/\|\mathbf{A}^\flat\|_\infty$ is upper bounded by an absolute constant, and so we have

$$\max_{\boldsymbol{y}' \in \mathcal{Y}}\langle \boldsymbol{x}, \mathbf{A}\boldsymbol{y}'\rangle - v \gtrsim P(\mathbf{A})\|\boldsymbol{x} - \boldsymbol{x}^\star\| = \frac{1}{2|N|\sqrt{|B|}}\sigma_{\min}(\overline{\mathbf{Q}})\left(1 + \frac{1}{\alpha_D(\mathbf{A})}\right)^{-1}\|\boldsymbol{x} - \boldsymbol{x}^\star\|$$

$$\gtrsim \frac{1}{|B|^3}(\alpha_D(\mathbf{A}))^2\gamma_P(\mathbf{A})\|\boldsymbol{x} - \boldsymbol{x}^\star\|.$$

Above, the last bound uses the fact that

$$\sigma_{\min}(\overline{\mathbf{Q}}) \geq \frac{1}{\sqrt{|\widetilde{B}|}}\min_{j \in \widetilde{N}}\text{dist}(\overline{\mathbf{Q}}_{:,j}, \text{span}(\overline{\mathbf{Q}}_{:,\widetilde{N}-j}))$$

$$\geq \frac{1}{|\widetilde{B}|^{3/2}}\min_{j \in \widetilde{N}}\text{dist}(\mathbf{Q}_{:,j}, \text{span}(\mathbf{Q}_{:,\widetilde{N}-j}))\alpha_D(\mathbf{A}) = \frac{1}{|\widetilde{B}|^{3/2}}\gamma_P(\mathbf{A})\alpha_D(\mathbf{A}),$$

where the first inequality uses (6), while the second one is a consequence of Lemma 3.4.

**Case II:** $\lambda P(\mathbf{A})\|\widehat{\boldsymbol{x}}_B - \boldsymbol{x}_B^\star\| < 4(1-\lambda)\|\mathbf{A}^\flat\|_\infty$. This case can only arise when $\overline{B} \neq \emptyset$ (for otherwise $\lambda = 1$). Then, we bound

$$
\begin{aligned}
\max_{\boldsymbol{y}' \in \mathcal{Y}} \langle \boldsymbol{x}, \mathbf{A}\boldsymbol{y}' \rangle - v &\geq \langle \boldsymbol{x}, \mathbf{A}\boldsymbol{y}^\star \rangle - v \\
&\geq \lambda(\langle \widehat{\boldsymbol{x}}_B - \boldsymbol{x}_B^\star, \mathbf{A}_{B,N}\boldsymbol{y}_N^\star \rangle) + (1-\lambda)(\langle \widehat{\boldsymbol{x}}_{\overline{B}}, \mathbf{A}_{\overline{B},N}\boldsymbol{y}_N^\star - v \rangle) \\
&\geq (1-\lambda)\beta_D(\mathbf{A}), \quad\quad\quad (29)
\end{aligned}
$$

by definition of $\beta_D(\mathbf{A})$ (Item 2) and the fact that $\langle \widehat{\boldsymbol{x}}_B - \boldsymbol{x}_B^\star, \mathbf{A}_{B,N}\boldsymbol{y}_N^\star \rangle = v\langle \widehat{\boldsymbol{x}}_B - \boldsymbol{x}_B^\star, \mathbf{1} \rangle = 0$. Moreover, by (27) together with the assumption that $\lambda P(\mathbf{A})\|\widehat{\boldsymbol{x}}_B - \boldsymbol{x}_B^\star\| < 4(1-\lambda)\|\mathbf{A}^\flat\|_\infty$,

$$
\|\boldsymbol{x} - \boldsymbol{x}^\star\|^2 \leq 32 \left( \frac{\|\mathbf{A}^\flat\|_\infty}{P(\mathbf{A})} \right)^2 (1-\lambda)^2 + 3(1-\lambda)^2 = \left( 32 \left( \frac{\|\mathbf{A}^\flat\|_\infty}{P(\mathbf{A})} \right)^2 + 3 \right)(1-\lambda)^2.
$$

Combining with (29) yields

$$
\begin{aligned}
\max_{\boldsymbol{y}' \in \mathcal{Y}} \langle \boldsymbol{x}, \mathbf{A}\boldsymbol{y}' \rangle - v &\geq \left( 32 \left( \frac{\|\mathbf{A}^\flat\|_\infty}{P(\mathbf{A})} \right)^2 + 3 \right)^{-2} \beta_D(\mathbf{A})\|\boldsymbol{x} - \boldsymbol{x}^\star\| \\
&\gtrsim \frac{P(\mathbf{A})}{\|\mathbf{A}^\flat\|_\infty} \beta_D(\mathbf{A})\|\boldsymbol{x} - \boldsymbol{x}^\star\| \\
&\gtrsim \frac{1}{\|\mathbf{A}^\flat\|_\infty} \frac{1}{|B|^3} \alpha_D(\mathbf{A})^2 \beta_D(\mathbf{A})\gamma_P(\mathbf{A})\|\boldsymbol{x} - \boldsymbol{x}^\star\|.
\end{aligned}
$$

$\square$

## C.2 Proofs from Section 3.3

We continue with the proofs from Section 3.3. As we have noted already, given that all quantities of interest in Definition 3.3 depend on the support of the equilibrium, it is natural to proceed by partitioning the probability space over all possible such configurations. To do so, we will use the following simple fact [Spielman and Teng, 2003, Proposition 8.1].

**Proposition C.5** (Spielman and Teng, 2003). *Let $X$ and $Y$ be random variables distributed according to an integrable density function. For any event $\mathcal{E}(X,Y)$,*

$$
\mathbb{P}_{X,Y}[\mathcal{E}(X,Y)] \leq \max_y \mathbb{P}_{X,Y}[\mathcal{E}(X,Y) \mid Y = y] =: \max_Y \mathbb{P}_{X,Y}[\mathcal{E}(X,Y) \mid Y].
$$

In our application, we want to condition on the event that $B$ is the support of $\boldsymbol{x}^\star$ and $N$ is the support of $\boldsymbol{y}^\star$. For convenience, we let $\mathsf{Type}_{B,N}(\mathbf{A})$ denote the indicator random variable representing whether $B$ and $N$ indeed index the positive coordinates of the equilibrium; that is, $\mathsf{Type}_{B,N}(\mathbf{A}) := \mathbb{1}\{B = \{i \in [n] : \boldsymbol{x}_i^\star(\mathbf{A}) > 0\} \wedge N = \{j \in [m] : \boldsymbol{y}_j^\star(\mathbf{A}) > 0\}\}$. Unlike general linear programs, which can be infeasible or unbounded, the linear program induced by a zero-sum game is guaranteed to be primal and dual feasible, no matter the perturbation (under Definition 1.1). We will thus only have to condition on events in which $B$ and $N$ are both nonempty. To be able to control the probability density function upon conditioning on $\mathsf{Type}_{B,N}(\mathbf{A})$, it will be convenient to perform a certain change of variables, which is described next.

**Change of variables** Let us denote by $\mathbf{A}_{\overline{B,N}}$ the entries of $\mathbf{A}$ excluding those in $\mathbf{A}_{B,N}$. We first perform a change of variables from $\mathbf{A}_{\overline{B,N}}, \mathbf{A}_{B,N}$ to $\mathbf{A}_{\overline{B,N}}, \mathbf{Q}, \boldsymbol{c}, \boldsymbol{b}, d$, which uses the linear transformation $\mathbf{T}$ associated with (5). With this new set of variables at hand, we can conveniently express $\mathbf{Q}\widetilde{\boldsymbol{y}}^\star = \boldsymbol{c}$ and $\mathbf{Q}^\top \widetilde{\boldsymbol{x}}^\star = \boldsymbol{b}$ (Claim C.3). Accordingly, we next perform a change of variables from $\mathbf{A}_{\overline{B,N}}, \mathbf{Q}, \boldsymbol{c}, \boldsymbol{b}, d$ to $\mathbf{A}_{\overline{B,N}}, \mathbf{Q}, \boldsymbol{x}^\star, \boldsymbol{y}^\star, v$. When performing those change of variables one has to account for the transformed probability density function. This can be understood as follows. The probability of an event $\mathcal{E}(\mathbf{A})$ can be expressed as

$$
\int_{\mathbf{A}} \mathcal{E}(\mathbf{A})\mu_{\mathbf{A}}(\mathbf{A}) d\mathbf{A}.
$$

The integral above can be cast in terms of a new set of variables $\mathbf{B}$ by computing the corresponding Jacobian, assuming that it is non-singular. We will make use of this fact in the sequel. The following lemma gathers some of the above observations regarding the change of variables.

**Lemma C.6** (Change of variables). *Let $\mathcal{E}(\mathbf{A})$ be any event that depends on the randomness of $\mathbf{A}$. Then,*

$$\mathbb{P}_{\mathbf{A}}[\mathcal{E}(\mathbf{A})] \leq \max_{B,N} \mathbb{P}_{\mathbf{A}}[\mathcal{E}(\mathbf{A}) \mid \mathsf{Type}_{B,N}(\mathbf{A})]$$

$$= \max_{B,N} \mathbb{P}_{\mathbf{A}_{\overline{B},N},\mathbf{Q},\boldsymbol{x}^\star,\boldsymbol{y}^\star,v}[\mathcal{E}(\mathbf{A}) \mid \mathbf{A}_{\overline{B},N}\boldsymbol{y}_N^\star \geq v\mathbf{1} \text{ and } \mathbf{A}_{N,B}^\top \boldsymbol{x}_B^\star \leq v\mathbf{1}].$$

Indeed, the first inequality above is a consequence of Proposition C.5. The equality then follows from noting that, when

$$\boldsymbol{c} = \mathbf{Q}\widetilde{\boldsymbol{y}}^\star, \boldsymbol{b} = \mathbf{Q}^\top \widetilde{\boldsymbol{x}}^\star, v = d - \langle \widetilde{\boldsymbol{x}}^\star, \mathbf{Q}\widetilde{\boldsymbol{y}}^\star\rangle \iff \mathbf{A}_{B,N}\boldsymbol{y}^\star = v\mathbf{1}, \mathbf{A}_{N,B}^\top \boldsymbol{x}^\star = v\mathbf{1},$$

the event $\mathsf{Type}_{B,N}(\mathbf{A})$ can be equivalently expressed as $\mathbf{A}_{\overline{B},N}\boldsymbol{y}_N^\star \geq v\mathbf{1}$ and $\mathbf{A}_{N,B}^\top \boldsymbol{x}_B^\star \leq v\mathbf{1}$.

We first bound the probability that $\beta_P(\mathbf{A}) \coloneqq \min_{j \in \overline{N}}(v - \langle \boldsymbol{x}_B^\star, \mathbf{A}_{B,j}\rangle)$ is close to 0; the proof for $\beta_D(\mathbf{A})$ is then symmetric. The key ingredient is the following anti-concentration lemma pertaining to a conditional Gaussian distribution [Spielman and Teng, 2003, Lemma 8.3].

**Lemma C.7** (Spielman and Teng, 2003). *Let $g$ be a Gaussian random variable of variance $\sigma^2$ and mean of absolute value at most 1. For $\epsilon \geq 0$, $\tau \geq 1$ and $t \leq \tau$,*

$$\mathbb{P}[g \leq t + \epsilon \mid g \geq t] \leq \frac{\epsilon\tau}{\sigma^2}e^{\frac{\epsilon(\tau+3)}{\sigma^2}}.$$

**Proposition 3.8.** *Let $\beta_P(\mathbf{A})$ be defined as in Item 2. For any $\epsilon \geq 0$,*

$$\mathbb{P}_{\mathbf{A}}\left[\beta_P(\mathbf{A}) \leq \frac{\epsilon}{5\|\mathbf{A}^\flat\|_\infty}\right] \leq \epsilon\frac{e\min(n,m)^2}{\sigma^2}.$$

*Proof.* By Lemma C.6, it suffices to bound

$$\max_{B,N} \mathbb{P}_{\mathbf{A}_{\overline{B},N},\mathbf{Q},\boldsymbol{x}^\star,\boldsymbol{y}^\star,v}[\beta_P(\mathbf{A}) \leq \epsilon' \mid \mathbf{A}_{\overline{B},N}\boldsymbol{y}_N^\star \geq v\mathbf{1} \text{ and } \mathbf{A}_{N,B}^\top \boldsymbol{x}_B^\star \leq v\mathbf{1}].$$

By Proposition C.5, it suffices to prove that for all $B, N, \mathbf{A}_{\overline{B},N}, \mathbf{A}_{\overline{B},\overline{N}}, \mathbf{Q}, \boldsymbol{x}^\star, \boldsymbol{y}^\star, v$ satisfying $\mathbf{A}_{\overline{B},N}\boldsymbol{y}_N^\star \geq v\mathbf{1}$,

$$\mathbb{P}_{\mathbf{A}_{B,\overline{N}}}[\exists j \in \overline{N} : v - \langle \boldsymbol{x}_B^\star, \mathbf{A}_{B,j}\rangle \leq \epsilon' \mid \forall j \in N : v - \langle \boldsymbol{x}_B^\star, \mathbf{A}_{B,j}\rangle \geq 0] \tag{30}$$

$$\leq \sum_{j \in \overline{N}} \mathbb{P}_{\mathbf{A}_{B,j}}[v - \langle \boldsymbol{x}_B^\star, \mathbf{A}_{B,j}\rangle \leq \epsilon' \mid \forall j \in N : v - \langle \boldsymbol{x}_B^\star, \mathbf{A}_{B,j}\rangle \geq 0] \tag{31}$$

$$\leq \sum_{j \in \overline{N}} \mathbb{P}_{\mathbf{A}_{B,j}}[v - \langle \boldsymbol{x}_B^\star, \mathbf{A}_{B,j}\rangle \leq \epsilon' \mid v - \langle \boldsymbol{x}_B^\star, \mathbf{A}_{B,j}\rangle \geq 0] \tag{32}$$

$$= \sum_{j \in \overline{N}} \mathbb{P}_{\boldsymbol{g}_j}[\boldsymbol{g}_j \leq \epsilon' - v \mid \boldsymbol{g}_j \geq -v]. \tag{33}$$

where in (30) the distribution of $\mathbf{A}_{B,\overline{N}}$ after conditioning on $\mathbf{A}_{\overline{B},N}, \mathbf{A}_{\overline{B},\overline{N}}, \mathbf{Q}, \boldsymbol{x}^\star, \boldsymbol{y}^\star, v$ remains the same, which is a consequence of independence per Definition 1.1; (31) is an application of the union bound; (32) uses the fact that the events $\{v - \langle \boldsymbol{x}_B^\star, \mathbf{A}_{B,j}\rangle \geq 0\}_{j \in N}$ are pairwise independent; and (33) defines $\boldsymbol{g}_j \coloneqq -\langle \boldsymbol{x}_B^\star, \mathbf{A}_{B,j}\rangle$, which is a Gaussian random variable with expectation $|\mathbb{E}[\boldsymbol{g}_j]| \leq \max_{i \in B}|\mathbf{A}_{i,j}|$ and variance $\mathbb{V}[\boldsymbol{g}_j] = \sum_{i \in B}(\boldsymbol{x}_i^\star)^2 \mathbb{V}[\mathbf{A}_{i,j}] = \sigma^2 \sum_{i \in B}(\boldsymbol{x}_i^\star)^2$ (by independence). In particular, by Cauchy-Schwarz, $\mathbb{V}[\boldsymbol{g}_j] \geq \frac{1}{|B|}\sigma^2$. Further, by Lemma C.7 (for $\tau = \max(1, |v|/|\mathbb{E}[\boldsymbol{g}_j]|)$), we have

$$\mathbb{P}_{\boldsymbol{g}_j}[\boldsymbol{g}_j \leq \epsilon' - v \mid \boldsymbol{g}_j \geq -v] \leq \epsilon'\frac{\max(|v|, |\mathbb{E}[\boldsymbol{g}_j]|)}{\mathbb{V}[\boldsymbol{g}_j]}e^{\epsilon'\frac{\max(4|\mathbb{E}[\boldsymbol{g}_j]|, 3|\mathbb{E}[\boldsymbol{g}_j]|+|v|)}{\mathbb{V}[\boldsymbol{g}_j]}}$$

$$\leq \epsilon'\frac{\min(n,m)\max(|v|, |\mathbb{E}[\boldsymbol{g}_j]|)}{\sigma^2}e^{\epsilon'\frac{\min(n,m)\max(4|\mathbb{E}[\boldsymbol{g}_j]|, 3|\mathbb{E}[\boldsymbol{g}_j]|+|v|)}{\sigma^2}}$$

for any $\epsilon' \geq 0$ and $j \in \overline{N}$, where we note that we applied Lemma C.7 for $\boldsymbol{g}_j/|\mathbb{E}[\boldsymbol{g}_j]|$ (since the absolute value of the mean has to be at most 1), which has variance $\mathbb{V}[\boldsymbol{g}_j]/(\mathbb{E}[\boldsymbol{g}_j])^2$. So, setting $\epsilon := \epsilon'(|v| + 4|\mathbb{E}[\boldsymbol{g}_j]|)$,

$$\mathop{\mathbb{P}}_{\boldsymbol{g}_j}\left[\boldsymbol{g}_j \leq \frac{\epsilon}{|v| + 4\max_{i \in B}|\mathbf{A}_{i,j}|} - v \mid \boldsymbol{g}_j \geq -v\right] \leq \mathop{\mathbb{P}}_{\boldsymbol{g}_j}\left[\boldsymbol{g}_j \leq \frac{\epsilon}{|v| + 4|\mathbb{E}[\boldsymbol{g}_j]|} - v \mid \boldsymbol{g}_j \geq -v\right]$$

$$\leq \epsilon\frac{\min(n,m)}{\sigma^2}e^{\epsilon\frac{\min(n,m)}{\sigma^2}}. \tag{34}$$

Now, when $\epsilon\frac{\min(n,m)}{\sigma^2} > 1$ the proposition is vacuously true, while in the contrary case the claim follows from (34) and (33). $\qquad\square$

Next, we proceed with the bound on $\gamma_P(\mathbf{A})$. The key ingredient is the observation that a random variable with a slowly changing density function cannot be too concentrated on any any interval (Lemma 3.7 due to Spielman and Teng [2003, Lemma 8.2]; we restate it below for convenience). Gaussian random variables have this property, as pointed out by Spielman and Teng [2003, Lemma 8.1].

**Lemma C.8** (Spielman and Teng, 2003). *Let $\mu$ be the probability density function of a Gaussian random variable in $\mathbb{R}^d$ of variance $\sigma^2$ centered at a point of norm at most 1. If $\mathrm{dist}(\boldsymbol{r}, \boldsymbol{r}') \leq \epsilon \leq 1$, then*

$$\frac{\mu(\boldsymbol{r}')}{\mu(\boldsymbol{r})} \geq e^{-\frac{\epsilon(\|\boldsymbol{r}\| + 2)}{\sigma^2}}.$$

**Lemma 3.7** (Spielman and Teng, 2003). *Let $\rho$ be the probability density function of a random variable $X$. If there exist $\delta > 0$ and $c \in (0, 1]$ such that*

$$0 \leq t \leq t' \leq \delta \implies \frac{\rho(t')}{\rho(t)} \geq c, \tag{7}$$

*then*

$$\mathbb{P}[X \leq \epsilon \mid X \geq 0] \leq \frac{\epsilon}{c\delta}.$$

**Proposition 3.9.** *Let $\gamma_P(\mathbf{A})$ be defined as in Item 3. For any $\epsilon \geq 0$,*

$$\mathop{\mathbb{P}}_{\mathbf{A}}\left[\gamma_P(\mathbf{A}) \leq \frac{\epsilon}{4\max_{j \in \widetilde{N}}\|\mathbf{Q}_{:,j}\| + 20\|\mathbf{A}^\flat\|_\infty + 3}\right] \leq \epsilon\frac{4e\min(n,m)^3}{\sigma^2}.$$

*Proof.* Let $\mu_{\mathbf{A}}(\mathbf{A})$ be the probability density function of $\mathbf{A}$, which, by independence (Definition 1.1), can be expressed as $\prod_{i \in [n], j \in [m]} \mu_{\mathbf{A}_{i,j}}$, where $\mu_{\mathbf{A}_{i,j}}$ is a Gaussian random variable. We first perform a change of variables from $\mathbf{A}_{\overline{B},N}, \mathbf{A}_{B,N}$ to $\mathbf{A}_{\overline{B},N}, \mathbf{Q}, \boldsymbol{b}, \boldsymbol{c}, d$, in accordance with (5); this can be understood through the (non-singular; Claim C.2) linear transformation $\mathbf{A}^\flat_{B,N} = \mathbf{T}(\mathbf{Q}^\flat, \boldsymbol{b}, \boldsymbol{c}, d)$. To express the density in the new variables, we first note that the Jacobian of the change of variables is $|\det(\mathbf{T})| = 1$ (Claim C.2), and so the density on $\mathbf{Q}, \boldsymbol{b}, \boldsymbol{c}, d$ can be expressed as $\mu_{\mathbf{A}_{B,N}}(\mathbf{T}(\mathbf{Q}^\flat, \boldsymbol{b}, \boldsymbol{c}, d))\mu_{\mathbf{A}_{\overline{B},N}}(\mathbf{A}_{\overline{B},N})$.

Next, we perform a change of variables from $\mathbf{A}_{\overline{B},N}, \mathbf{Q}, \boldsymbol{b}, \boldsymbol{c}, d$ to $\mathbf{A}_{\overline{B},N}, \mathbf{Q}, \widetilde{\boldsymbol{x}}^\star, \widetilde{\boldsymbol{y}}^\star, v$ according to the transformations $\mathbf{Q}\widetilde{\boldsymbol{y}}^\star = \boldsymbol{c}$; $\mathbf{Q}^\top\widetilde{\boldsymbol{x}}^\star = \boldsymbol{b}$; and $v = d - \langle\widetilde{\boldsymbol{x}}^\star, \mathbf{Q}\widetilde{\boldsymbol{y}}^\star\rangle$. It is easy to see that the Jacobian of the change of variables is

$$\left|\det\left(\frac{\partial(\mathbf{A}_{\overline{B},N}, \mathbf{Q}, \boldsymbol{b}, \boldsymbol{c}, d)}{\partial(\mathbf{A}_{\overline{B},N}, \mathbf{Q}, \widetilde{\boldsymbol{x}}^\star, \widetilde{\boldsymbol{y}}^\star, v)}\right)\right| = \left|\det\left(\frac{\partial(\boldsymbol{b}, \boldsymbol{c}, d)}{\partial(\widetilde{\boldsymbol{x}}^\star, \widetilde{\boldsymbol{y}}^\star, v)}\right)\right| = \det(\mathbf{Q})^2.$$

So, the density on $\mathbf{A}_{\overline{B},N}, \mathbf{Q}, \widetilde{\boldsymbol{x}}^\star, \widetilde{\boldsymbol{y}}^\star, v$ reads

$$\mu_{\mathbf{A}_{B,N}}(\mathbf{T}(\mathbf{Q}^\flat, \mathbf{Q}^\top\widetilde{\boldsymbol{x}}^\star, \mathbf{Q}\widetilde{\boldsymbol{y}}^\star, v + \langle\widetilde{\boldsymbol{x}}^\star, \mathbf{Q}\widetilde{\boldsymbol{y}}^\star\rangle))\mu_{\mathbf{A}_{\overline{B},N}}(\mathbf{A}_{\overline{B},N})\det(\mathbf{Q})^2.$$

By Lemma C.6, it suffices to upper bound

$$\max_{B,N}\mathop{\mathbb{P}}_{\mathbf{A}_{\overline{B},N}, \mathbf{Q}, \boldsymbol{x}^\star, \boldsymbol{y}^\star, v}[\gamma_P(\mathbf{A}) \leq \epsilon \mid \mathbf{A}_{\overline{B},N}\boldsymbol{y}^\star_N \geq v\mathbf{1} \text{ and } \mathbf{A}^\top_{N,B}\boldsymbol{x}^\star_B \leq v\mathbf{1}].$$

Further, by Proposition C.5, it is in turn enough to bound $\mathbb{P}_{\mathbf{Q}}[\gamma_P(\mathbf{A}) \leq \epsilon]$ for all $B$, $N$ (for the non-trivial case where $\widetilde{B}, \widetilde{N} \neq \emptyset$), $\mathbf{A}_{\overline{B},N}$, $\widetilde{\boldsymbol{x}}^{\star}$, $\widetilde{\boldsymbol{y}}^{\star}$, $v$ such that $\mathbf{A}_{\overline{B},N}\boldsymbol{y}_N^{\star} \geq v\mathbf{1}$ and $\mathbf{A}_{\overline{N},B}^{\top}\boldsymbol{x}_B^{\star} \leq v\mathbf{1}$, where the induced distribution on $\mathbf{Q}$ is

$$\mu_{\mathbf{A}_{B,N}}(\mathbf{T}(\mathbf{Q}^{\flat}, \mathbf{Q}^{\top}\widetilde{\boldsymbol{x}}^{\star}, \mathbf{Q}\widetilde{\boldsymbol{y}}^{\star}, v + \langle\widetilde{\boldsymbol{x}}^{\star}, \mathbf{Q}\widetilde{\boldsymbol{y}}^{\star}\rangle)) \det(\mathbf{Q})^2.$$

We will prove that for any $j \in \widetilde{N}$ and $\mathbf{Q}_{:,\widetilde{N}-j}$,

$$\mathbb{P}_{\mathbf{Q}_{:,j}}\left[\mathsf{dist}(\mathbf{Q}_{:,j}, \mathsf{span}(\mathbf{Q}_{:,\widetilde{N}-j})) \leq \frac{\epsilon}{4\|\mathbf{Q}_{:,j}\| + 4|v| + 4\|\mathbf{Q}_{:,\widetilde{N}-j}^{\flat}\|_{\infty} + 3}\right] \leq \epsilon \frac{4e\min(n,m)^2}{\sigma^2}, \quad (35)$$

and then apply a union bound over $j \in \widetilde{N}$. Having fixed $\mathbf{Q}_{:,\widetilde{N}-j}$, we can express $\mathbf{Q}_{:,j}$ as $\boldsymbol{q}^{\|} + t\boldsymbol{q}^{\perp}$, where $\mathbb{R}^{\widetilde{B}} \ni \boldsymbol{q}^{\|} \in \mathsf{span}(\mathbf{Q}_{:,\widetilde{N}-j})$ and $\mathbb{R}^{\widetilde{B}} \ni \boldsymbol{q}^{\perp}$ is the unit vector orthogonal to $\mathsf{span}(\mathbf{Q}_{:,\widetilde{N}-j})$. Then, $|t| = \mathsf{dist}(\mathbf{Q}_{:,j}, \mathsf{span}(\mathbf{Q}_{:,\widetilde{N}-j}))$ and $|\det(\mathbf{Q})| = tC(\mathbf{Q}_{:,\widetilde{N}-j})$, where $C(\mathbf{Q}_{:,\widetilde{N}-j})$ does not depend on $\mathbf{Q}_{:,j}$ (this can be obtained by expressing the determinant using the formula for parallelepipeds). By symmetry, we can prove (35) by bounding the probability that $t$ is at most $\epsilon$ given that $t$ is at least $0$. We can thus focus on proving

$$\max_{\boldsymbol{q}^{\|} \in \mathsf{span}(\mathbf{Q}_{:,\widetilde{N}-j})} \mathbb{P}_t[t \leq \epsilon \mid t \geq 0] \leq \epsilon \frac{4e\min(n,m)^2(4\|\boldsymbol{q}^{\|}\|_{\infty} + 4|v| + 4\|\mathbf{Q}_{:,\widetilde{N}-j}^{\flat}\|_{\infty} + 3)}{\sigma^2}, \quad (36)$$

and then (35) follows from the fact that $\|\mathbf{Q}_{:,j}\| \geq \|\boldsymbol{q}^{\|}\|$. Now, the induced distribution on $t$ is proportional to

$$\rho(t) := t^2 \prod_{(i,j)\in B\times N} \mu_{\mathbf{A}_{i,j}}(\langle\mathbf{T}_{i,j}, \boldsymbol{r}_{i,j}(t)\rangle)$$

for $\boldsymbol{r}_{i,j}(t)$ defined as

$$(\boldsymbol{q}^{\|} + t\boldsymbol{q}^{\perp}, \mathbf{Q}_{:,\widetilde{N}-j}^{\flat}, \mathbf{Q}_{\widetilde{N}-j,:}^{\top}\widetilde{\boldsymbol{x}}^{\star}, \langle\widetilde{\boldsymbol{x}}^{\star}, \boldsymbol{q}^{\|} + t\boldsymbol{q}^{\perp}\rangle, \mathbf{Q}_{:,\widetilde{N}-j}\widetilde{\boldsymbol{y}}_{\widetilde{N}-j}^{\star} + \widetilde{\boldsymbol{y}}_j^{\star}(\boldsymbol{q}^{\|} + t\boldsymbol{q}^{\perp}),$$
$$v + \langle\widetilde{\boldsymbol{x}}^{\star}, \mathbf{Q}_{:,\widetilde{N}-j}\widetilde{\boldsymbol{y}}_{\widetilde{N}-j}^{\star}\rangle + \widetilde{\boldsymbol{y}}_j^{\star}\langle\widetilde{\boldsymbol{x}}^{\star}, \boldsymbol{q}^{\|} + t\boldsymbol{q}^{\perp}\rangle).$$

We now want to apply Lemma 3.7. To that end, we have

$$|\langle\mathbf{T}_{i,j}, \boldsymbol{r}_{i,j}(t) - \boldsymbol{r}_{i,j}(t')\rangle|^2 \leq \|\mathbf{T}_{i,j}\|^2\|\boldsymbol{r}_{i,j}(t) - \boldsymbol{r}_{i,j}(t')\|^2$$
$$\leq 4(t-t')^2\|(\boldsymbol{q}^{\perp}, \langle\widetilde{\boldsymbol{x}}^{\star}, \boldsymbol{q}^{\perp}\rangle, \widetilde{\boldsymbol{y}}_j^{\star}\boldsymbol{q}^{\perp}, \widetilde{\boldsymbol{y}}_j^{\star}\langle\widetilde{\boldsymbol{x}}^{\star}, \boldsymbol{q}^{\perp}\rangle)\|^2 \quad (37)$$
$$\leq 16(t-t')^2, \quad (38)$$

where (37) follows from $\|\mathbf{T}_{i,j}\|_2 \leq 2$ (Claim C.2), and (38) follows from the fact that $\|\boldsymbol{q}^{\perp}\|, \|\widetilde{\boldsymbol{x}}^{\star}\|, \|\widetilde{\boldsymbol{y}}^{\star}\| \leq 1$. Moreover, again by Claim C.2,

$$|\langle\mathbf{T}_{i,j}, \boldsymbol{r}_{i,j}(t)\rangle| \leq \|\mathbf{T}_{i,j}\|_1\|\boldsymbol{r}_{i,j}(t)\|_{\infty} \leq 4(\|\boldsymbol{q}^{\|}\|_{\infty} + |v| + \|\mathbf{Q}_{:,\widetilde{N}-j}^{\flat}\|_{\infty} + t).$$

Let $0 \leq t \leq t' \leq \delta \leq \frac{1}{4}$ for $\delta = \frac{\sigma^2}{4|B||N|(4\|\boldsymbol{q}^{\|}\|+4|v|+4\|\mathbf{Q}_{:,\widetilde{N}-j}^{\flat}\|_{\infty}+3)}$. Lemma C.8 then implies that

$$\frac{\mu_{\mathbf{A}_{i,j}}(\langle\mathbf{T}_{i,j}, \boldsymbol{r}_{i,j}(t')\rangle)}{\mu_{\mathbf{A}_{i,j}}(\langle\mathbf{T}_{i,j}, \boldsymbol{r}_{i,j}(t)\rangle)} \geq e^{-\frac{1}{|B||N|}}.$$

Thus,

$$\frac{\rho(t')}{\rho(t)} \geq \left(\frac{t'}{t}\right)^2 \prod_{(i,j)\in B\times N} \frac{\mu_{\mathbf{A}_{i,j}}(\langle\mathbf{T}_{i,j}, \boldsymbol{r}_{i,j}(t')\rangle)}{\mu_{\mathbf{A}_{i,j}}(\langle\mathbf{T}_{i,j}, \boldsymbol{r}_{i,j}(t)\rangle)} \geq e^{-1}.$$

We conclude that (36) can be obtained from Lemma 3.7, and the theorem follows. $\square$

Finally, we bound the probability that $\alpha_P(\mathbf{A})$ (Item 1) is close to 0; $\alpha_D(\mathbf{A})$ can be bounded in a similar fashion.

**Proposition 3.10.** *Let $\alpha_P(\mathbf{A})$ be defined as in [Item 1]. For any $\epsilon \geq 0$,*

$$\mathbb{P}_{\mathbf{A}}\left[\alpha_P(\mathbf{A}) \leq \frac{\epsilon}{25(\|\mathbf{A}^\flat\|_\infty + 1)^2}\right] \leq \epsilon \frac{8e^2 mn \min(n,m)}{\sigma^2}.$$

*Proof.* By [Lemma C.6], it suffices to bound

$$\max_{B,N} \mathbb{P}_{\mathbf{A}_{\overline{B},\overline{N}},\mathbf{Q},\boldsymbol{x}^\star,\boldsymbol{y}^\star,v}[\alpha_P(\mathbf{A}) \leq \epsilon \mid \mathbf{A}_{\overline{B},N}\boldsymbol{y}_N^\star \geq v\mathbf{1} \text{ and } \mathbf{A}_{\overline{N},B}^\top \boldsymbol{x}_B^\star \leq v\mathbf{1}],$$

where we recall that the induced probability density function on $\mathbf{A}_{\overline{B},\overline{N}}, \mathbf{Q}, \boldsymbol{x}^\star, \boldsymbol{y}^\star, v$ reads

$$\mu_{\mathbf{A}_{B,N}}(\mathbf{T}(\mathbf{Q}^\flat, \mathbf{Q}^\top \widetilde{\boldsymbol{x}}^\star, \mathbf{Q}\widetilde{\boldsymbol{y}}^\star, v + \langle \widetilde{\boldsymbol{x}}^\star, \mathbf{Q}\widetilde{\boldsymbol{y}}^\star \rangle))\mu_{\mathbf{A}_{B,\overline{N}}}(\mathbf{A}_{B,\overline{N}})\mu_{\mathbf{A}_{\overline{B},N}}(\mathbf{A}_{\overline{B},N})\mu_{\mathbf{A}_{\overline{B},\overline{N}}}(\mathbf{A}_{\overline{B},\overline{N}})\det(\mathbf{Q})^2.$$

We consider the non-trivial case where $\widetilde{B}, \widetilde{N} \neq \emptyset$. We will perform a further change of variables. Namely, let $\boldsymbol{a} = \mathbf{A}_{\overline{N},i}$ for $i \in B \setminus \widetilde{B}$. We map $\mathbf{A}_{B,\overline{N}}$ to $\overline{\mathbf{A}}_{\widetilde{B},\overline{N}} := \mathbf{A}_{\widetilde{B},\overline{N}} - \mathbf{1}\boldsymbol{a}^\top, \boldsymbol{a}$, so that $\mathbf{A}_{\overline{N},B}^\top \boldsymbol{x}_B^\star \leq v\mathbf{1}$ can be equivalently expressed as $\overline{\mathbf{A}}_{\overline{N},\widetilde{B}}^\top \widetilde{\boldsymbol{x}}^\star \leq v\mathbf{1} - \boldsymbol{a}$. The induced density function is now proportional to

$$\mu_{\mathbf{A}_{B,N}}(\mathbf{T}(\mathbf{Q}^\flat, \mathbf{Q}^\top \widetilde{\boldsymbol{x}}^\star, \mathbf{Q}\widetilde{\boldsymbol{y}}^\star, v + \langle \widetilde{\boldsymbol{x}}^\star, \mathbf{Q}\widetilde{\boldsymbol{y}}^\star \rangle))\mu_{\boldsymbol{a}}(\boldsymbol{a})\mu_{\mathbf{A}_{\tilde{B},\overline{N}}}(\overline{\mathbf{A}}_{\widetilde{B},\overline{N}} + \mathbf{1}\boldsymbol{a}^\top)\nu(\cdot),$$

where $\nu(\cdot)$ does not depend on $\widetilde{\boldsymbol{x}}^\star$ and $\boldsymbol{a}$. By [Proposition C.5], it is enough to show that for any $B, N, \overline{\mathbf{A}}_{\widetilde{B},\overline{N}}, \mathbf{A}_{\overline{B},N}, \mathbf{A}_{\overline{B},\overline{N}}, \mathbf{Q}, \boldsymbol{y}^\star, v$ satisfying $\mathbf{A}_{\overline{B},N}\boldsymbol{y}^\star \geq v\mathbf{1}$,

$$\mathbb{P}_{\widetilde{\boldsymbol{x}}^\star,\boldsymbol{a}}\left[\alpha_P \leq \frac{\epsilon}{\max((\|\mathbf{Q}^\flat\|_\infty + 1)^2, (1 + \|\overline{\mathbf{A}}_{\widetilde{B},\overline{N}}^\flat\|_\infty)(5\|\overline{\mathbf{A}}_{\widetilde{B},\overline{N}}^\flat\|_\infty + |v| + 4))} \mid \overline{\mathbf{A}}_{\overline{N},\widetilde{B}}^\top \widetilde{\boldsymbol{x}}^\star \leq v\mathbf{1} - \boldsymbol{a}\right]$$

$$\leq \epsilon \frac{8e^2 mn \min(n,m)}{\sigma^2},$$

where the induced distribution on $\widetilde{\boldsymbol{x}}^\star$ and $\boldsymbol{a}$ is proportional to

$$\mu_{\mathbf{A}_{B,N}}(\mathbf{T}(\mathbf{Q}^\flat, \mathbf{Q}^\top \widetilde{\boldsymbol{x}}^\star, \mathbf{Q}\widetilde{\boldsymbol{y}}^\star, v + \langle \widetilde{\boldsymbol{x}}^\star, \mathbf{Q}\widetilde{\boldsymbol{y}}^\star \rangle))\mu_{\boldsymbol{a}}(\boldsymbol{a})\mu_{\mathbf{A}_{\tilde{B},\overline{N}}}(\overline{\mathbf{A}}_{\widetilde{B},\overline{N}} + \mathbf{1}\boldsymbol{a}^\top). \tag{39}$$

We see that $\widetilde{\boldsymbol{x}}^\star$ is independent of $\boldsymbol{a}$ and $\{\boldsymbol{a}_j\}_{j \in \overline{N}}$ are pairwise independent. Thus, conditioning on the event $\overline{\mathbf{A}}_{\overline{N},\widetilde{B}}^\top \widetilde{\boldsymbol{x}}^\star \leq v\mathbf{1} - \boldsymbol{a}$, the induced distribution on $\widetilde{\boldsymbol{x}}^\star$ is proportional to

$$\mu_{\mathbf{A}_{B,N}}(\mathbf{T}(\mathbf{Q}^\flat, \mathbf{Q}^\top \widetilde{\boldsymbol{x}}^\star, \mathbf{Q}\widetilde{\boldsymbol{y}}^\star, v + \langle \widetilde{\boldsymbol{x}}^\star, \mathbf{Q}\widetilde{\boldsymbol{y}}^\star \rangle)) \prod_{j \in \overline{N}} \mathbb{P}_{\boldsymbol{a}_j}[\langle \overline{\mathbf{A}}_{\widetilde{B},j}, \widetilde{\boldsymbol{x}}^\star \rangle \leq v - \boldsymbol{a}_j].$$

We can proceed by showing that for any fixed $i \in \widetilde{B}$ and $\widetilde{\boldsymbol{x}}_{\widetilde{B}-i}^\star$,

$$\mathbb{P}_{\widetilde{\boldsymbol{x}}_i^\star}\left[\widetilde{\boldsymbol{x}}_i^\star \leq \frac{\epsilon}{\max((\|\mathbf{Q}^\flat\|_\infty + 1)^2, (1 + \|\overline{\mathbf{A}}_{\widetilde{B},\overline{N}}^\flat\|_\infty)(5\|\overline{\mathbf{A}}_{\widetilde{B},\overline{N}}^\flat\|_\infty + |v| + 4))} \mid \overline{\mathbf{A}}_{\overline{N},\widetilde{B}}^\top \widetilde{\boldsymbol{x}}^\star \leq v\mathbf{1} - \boldsymbol{a}\right]$$

$$\leq \epsilon \frac{8e^2 m \min(n,m)}{\sigma^2},$$

and then applying the union bound over all $i \in \widetilde{B}$. Having fixed $\widetilde{\boldsymbol{x}}_{\widetilde{B}-i}^\star$, the induced density on $\widetilde{\boldsymbol{x}}_i^\star$, say $\rho(t)$, is proportional to $\rho_1(t) \cdot \rho_2(t)$, where

$$\rho_1(t) := \mu_{\mathbf{A}_{B,N}}(\mathbf{T}(\mathbf{Q}^\flat, \mathbf{Q}_{:,\widetilde{B}-i}^\top \widetilde{\boldsymbol{x}}_{\widetilde{B}-i}^\star + t\mathbf{Q}_{:,i}^\top, \mathbf{Q}\widetilde{\boldsymbol{y}}^\star, v + \langle \widetilde{\boldsymbol{x}}_{\widetilde{B}-i}^\star, \mathbf{Q}_{\widetilde{B}-i,:}\widetilde{\boldsymbol{y}}^\star \rangle + t\langle \mathbf{Q}_{i,:}, \widetilde{\boldsymbol{y}}^\star \rangle))$$

and

$$\rho_2(t) := \prod_{j \in \overline{N}} \mathbb{P}_{\boldsymbol{a}_j}[\langle \overline{\mathbf{A}}_{j,\widetilde{B}-i}, \widetilde{\boldsymbol{x}}_{\widetilde{B}-i}^\star \rangle + \overline{\mathbf{A}}_{i,j}t \leq v - \boldsymbol{a}_j].$$

We will first apply [Lemma 3.7] to bound $\rho_1(t')/\rho_1(t)$ for $0 \leq t \leq t' \leq \delta \leq 1$ and a sufficiently small $\delta$. We define

$$\boldsymbol{r}_{i,j}(t) := (\mathbf{Q}^\flat, \mathbf{Q}_{:,\widetilde{B}-i}^\top \widetilde{\boldsymbol{x}}_{\widetilde{B}-i}^\star + t\mathbf{Q}_{:,i}^\top, \mathbf{Q}\widetilde{\boldsymbol{y}}^\star, v + \langle \widetilde{\boldsymbol{x}}_{\widetilde{B}-i}^\star, \mathbf{Q}_{\widetilde{B}-i,:}\widetilde{\boldsymbol{y}}^\star \rangle + t\langle \mathbf{Q}_{i,:}, \widetilde{\boldsymbol{y}}^\star \rangle),$$

so that $\rho_1(t) = \prod_{(i,j) \in B \times N} \mu_{\mathbf{A}_{i,j}}(\langle \mathbf{T}_{i,j}, \boldsymbol{r}_{i,j}(t) \rangle)$. Then, we have

$$|\langle \mathbf{T}_{i,j}, \boldsymbol{r}_{i,j}(t) - \boldsymbol{r}_{i,j}(t') \rangle| \le 4|t - t'| \|\mathbf{Q}^\flat\|_\infty,$$

where we used Claim C.2. Further,

$$|\langle \mathbf{T}_{i,j}, \boldsymbol{r}_{i,j}(t) \rangle| \le (t+1) \|\mathbf{Q}^\flat\|_\infty,$$

and so Lemma C.8 implies that for $\delta \le \frac{1}{4\|\mathbf{Q}^\flat\|_\infty}$,

$$\frac{\mu_{\mathbf{A}_{i,j}}(\langle \mathbf{T}_{i,j}, \boldsymbol{r}_{i,j}(t') \rangle)}{\mu_{\mathbf{A}_{i,j}}(\langle \mathbf{T}_{i,j}, \boldsymbol{r}_{i,j}(t) \rangle)} \ge e^{-\frac{8\delta\|\mathbf{Q}^\flat\|_\infty(\|\mathbf{Q}^\flat\|_\infty+1)}{\sigma^2}}.$$

As a result, for $\delta \le \frac{\sigma^2}{8|B||N|\|\mathbf{Q}^\flat\|_\infty(\|\mathbf{Q}^\flat\|_\infty+1)}$,

$$\frac{\rho_1(t')}{\rho_1(t)} = \prod_{(i,j) \in B \times N} \frac{\mu_{\mathbf{A}_{i,j}}(\langle \mathbf{T}_{i,j}, \boldsymbol{r}_{i,j}(t') \rangle)}{\mu_{\mathbf{A}_{i,j}}(\langle \mathbf{T}_{i,j}, \boldsymbol{r}_{i,j}(t) \rangle)} \ge e^{-1}.$$

Next, we focus on lower bounding $\rho_2(t')/\rho_2(t)$. From (39), it is not hard to see that $\boldsymbol{a}_j$ is a Gaussian random variable with expectation $|\mathbb{E}[\boldsymbol{a}_j]| \le 1 + \|\overline{\mathbf{A}}^\flat_{\widetilde{B},\overline{N}}\|_\infty$ and variance $\mathbb{V}[\boldsymbol{a}_j] \ge \frac{\sigma^2}{\min(n,m)}$. Also,

$$\begin{aligned}
\frac{\rho_2(t')}{\rho_2(t)} &= \prod_{j \in \overline{N}} \frac{\mathbb{P}_{\boldsymbol{a}_j}[\langle \overline{\mathbf{A}}_{\widetilde{B}-i,j}, \widetilde{\boldsymbol{x}}^\star_{\widetilde{B}-i} \rangle + \overline{\mathbf{A}}_{i,j}t' \le v - \boldsymbol{a}_j]}{\mathbb{P}_{\boldsymbol{a}_j}[\langle \overline{\mathbf{A}}_{\widetilde{B}-i,j}, \widetilde{\boldsymbol{x}}^\star_{\widetilde{B}-i} \rangle + \overline{\mathbf{A}}_{i,j}t \le v - \boldsymbol{a}_j]} \\
&\ge \prod_{j \in \overline{N}} \mathbb{P}_{\boldsymbol{a}_j}[\langle \overline{\mathbf{A}}_{\widetilde{B}-i,j}, \widetilde{\boldsymbol{x}}^\star_{\widetilde{B}-i} \rangle + \overline{\mathbf{A}}_{i,j}t' \le v - \boldsymbol{a}_j \mid \langle \overline{\mathbf{A}}_{\widetilde{B}-i,j}, \widetilde{\boldsymbol{x}}^\star_{\widetilde{B}-i} \rangle + \overline{\mathbf{A}}_{i,j}t \le v - \boldsymbol{a}_j].
\end{aligned}$$

By Lemma C.7 (for $\tau = (2\|\overline{\mathbf{A}}^\flat_{\widetilde{B},\overline{N}}\|_\infty + |v| + 1)/(1 + \|\overline{\mathbf{A}}^\flat_{\widetilde{B},\overline{N}}\|_\infty)$),

$$\begin{aligned}
&\mathbb{P}_{\boldsymbol{a}_j}[\langle \overline{\mathbf{A}}_{\widetilde{B}-i,j}, \widetilde{\boldsymbol{x}}^\star_{\widetilde{B}-i} \rangle + \overline{\mathbf{A}}_{i,j}t' \le v - \boldsymbol{a}_j \mid \langle \overline{\mathbf{A}}_{\widetilde{B}-i,j}, \widetilde{\boldsymbol{x}}^\star_{\widetilde{B}-i} \rangle + \overline{\mathbf{A}}_{i,j}t \le v - \boldsymbol{a}_j] \\
&\ge 1 - \delta\frac{\min(n,m)\|\overline{\mathbf{A}}^\flat_{\widetilde{B},\overline{N}}\|_\infty(2\|\overline{\mathbf{A}}^\flat_{\widetilde{B},\overline{N}}\|_\infty + |v| + 1)}{\sigma^2} e^{\delta\frac{\min(n,m)\|\overline{\mathbf{A}}^\flat_{\widetilde{B},\overline{N}}\|_\infty(5\|\overline{\mathbf{A}}^\flat_{\widetilde{B},\overline{N}}\|_\infty + |v| + 4)}{\sigma^2}}.
\end{aligned}$$

Thus, for $\delta \le \frac{1}{2em}\frac{\sigma^2}{\min(n,m)\|\overline{\mathbf{A}}^\flat_{\widetilde{B},\overline{N}}\|_\infty(5\|\overline{\mathbf{A}}^\flat_{\widetilde{B},\overline{N}}\|_\infty + |v| + 4)}$,

$$\mathbb{P}_{\boldsymbol{a}_j}[\langle \overline{\mathbf{A}}_{\widetilde{B}-i,j}, \widetilde{\boldsymbol{x}}^\star_{\widetilde{B}-i} \rangle + \overline{\mathbf{A}}_{i,j}t' \le v - \boldsymbol{a}_j \mid \langle \overline{\mathbf{A}}_{\widetilde{B}-i,j}, \widetilde{\boldsymbol{x}}^\star_{\widetilde{B}-i} \rangle + \overline{\mathbf{A}}_{i,j}t \le v - \boldsymbol{a}_j] \ge 1 - \frac{1}{2m},$$

which in turn implies that

$$\frac{\rho_2(t')}{\rho_2(t)} \ge \left(1 - \frac{1}{2m}\right)^{\overline{N}} \ge e^{-1}.$$

We conclude that $\frac{\rho(t')}{\rho(t)} \ge e^{-2}$, and the proof follows from Lemma 3.7 by lower bounding the value of $\delta$. $\qquad\square$

Armed with Propositions 3.8 to 3.10, Theorem 1.4 can be obtained from Theorem 3.6, in conjunction with a union bound and the fact that $\|\mathbf{A}^\flat\|_\infty \le \mathsf{poly}(n,m)$ with high probability (by Gaussian concentration).

## C.3  Proof of Theorem 1.2

Having established Theorem 1.4, here we explain how existing results imply Theorem 1.2. We first focus on OGDA. We also take the opportunity to explain in more detail how Wei et al. [2021] established Definition 1.3, which was sketched earlier in Section 3.1. Our treatment of the rest of the algorithms will be more brief.

**Metric subregularity**   A central ingredient in the approach of Wei et al. [2021] is what they refer to as saddle-point *metric subregularity*, stated below as Definition C.9. For the sake of generality, we give the definition for a general objective function $f : \mathcal{X} \times \mathcal{Y} \ni (\boldsymbol{x}, \boldsymbol{y}) \mapsto f(\boldsymbol{x}, \boldsymbol{y})$, assumed to be continuously differentiable; (1) corresponds to the bilinear case $f(\boldsymbol{x}, \boldsymbol{y}) = \langle \boldsymbol{x}, \mathbf{A}\boldsymbol{y} \rangle$. We use again the notation $F(\boldsymbol{z}) := (\nabla_{\boldsymbol{x}} f(\boldsymbol{x}, \boldsymbol{y}), -\nabla_{\boldsymbol{y}} f(\boldsymbol{x}, \boldsymbol{y}))$, where $\mathbb{R}^{n+m} \ni \boldsymbol{z} := (\boldsymbol{x}, \boldsymbol{y})$. We also let $L \in \mathbb{R}_{>0}$ be a Lipschitz continuity parameter for $F$ with respect to $\| \cdot \|$, so that $\|F(\boldsymbol{z}) - F(\boldsymbol{z}')\| \leq L\|\boldsymbol{z} - \boldsymbol{z}'\|$; in the context of (1), one can always take $L := \|\mathbf{A}\|$.

**Definition C.9** (Metric subregularity for saddle-point problems [Wei et al., 2021])**.**   A saddle-point problem satisfies *metric subregularity* if there exists a problem-dependent parameter $\kappa' \in \mathbb{R}_{>0}$ such that for any $\boldsymbol{z} \in \mathcal{Z}$ and $\boldsymbol{z}^{\star} := \Pi_{\mathcal{Z}^{\star}}(\boldsymbol{z})$,

$$\sup_{\boldsymbol{z}' \in \mathcal{Z}} \frac{\langle F(\boldsymbol{z}), \boldsymbol{z} - \boldsymbol{z}' \rangle}{\|\boldsymbol{z} - \boldsymbol{z}'\|} \geq \kappa'\|\boldsymbol{z} - \boldsymbol{z}^{\star}\|. \tag{40}$$

The nomenclature of Definition C.9 can be justified by the fact that (40) is equivalent to a common type of metric subregularity [Wei et al., 2021, Appendix F]; for more background, we refer to Dontchev and Rockafellar [2009]. We further remark that Wei et al. [2021] introduced (40) in a more general form by allowing an exponent $\beta \in \mathbb{R}_{\geq 0}$ in the right-hand side, but that additional flexibility is not relevant for our purposes.[6]

Now, there an obvious connection between Definition 1.3 and Definition C.9 in bilinear problems with bounded domain; namely, we have

$$\sup_{\boldsymbol{z}' \in \mathcal{Z}} \frac{\langle F(\boldsymbol{z}), \boldsymbol{z} - \boldsymbol{z}' \rangle}{\|\boldsymbol{z} - \boldsymbol{z}'\|} \geq \frac{1}{2}\Phi(\boldsymbol{z}),$$

where we used the fact that $\langle F(\boldsymbol{z}), \boldsymbol{z} \rangle = 0$ and $\|\boldsymbol{z} - \boldsymbol{z}'\| \leq D_{\mathcal{Z}} = 2$. So, Definition 1.3 with respect to parameter $\kappa$ implies Definition C.9 with parameter $\kappa' := \kappa/2$.

**Linear convergence of `OGDA`**   Under metric subregularity, in the sense of Definition C.9, Wei et al. [2021] were able to establish that `OGDA` converges to the set $\mathcal{Z}^{\star}$ at a linear rate:

**Theorem C.10** (Wei et al., 2021)**.**   *Consider a saddle-point problem* (1) *satisfying metric subregularity with respect to some* $\kappa' \in \mathbb{R}_{>0}$. *For any* $\eta \leq \frac{1}{8L}$, *the iterates* $(\boldsymbol{z}^{(\tau)})_{1 \leq \tau \leq t}$ *of* `OGDA` *satisfy*

$$\mathsf{dist}(\boldsymbol{z}^{(t)}, \mathcal{Z}^{\star}) \leq 8 \left( 1 + \frac{16\eta^2(\kappa')^2}{81} \right)^{-t/2} \mathsf{dist}(\widehat{\boldsymbol{z}}^{(1)}, \mathcal{Z}^{\star}). \tag{41}$$

As a result, Theorem C.10 implies that `OGDA` guarantees $\mathsf{dist}(\boldsymbol{z}^{(t)}, \mathcal{Z}^{\star}) \leq \epsilon$ so long as

$$t \geq 2 \left\lceil \frac{\log\left(\frac{8D_{\mathcal{Z}}}{\epsilon}\right)}{\log\left(1 + \frac{(\kappa')^2}{324\|\mathbf{A}\|^2}\right)} \right\rceil. \tag{42}$$

In conjunction with Theorem 3.6 and Propositions 3.8 to 3.10, this immediately implies that `OGDA` has a polynomial smoothed complexity with high probability, as claimed earlier in Theorem 1.2.

Before we proceed, it is instructive to explain how Wei et al. [2021] treated the error bound in bilinear problems where $\mathcal{X}$ and $\mathcal{Y}$ are polyhedral sets. As we explained earlier, it is enough to show that for any $\boldsymbol{x} \in \mathcal{X}$ and $\boldsymbol{y} \in \mathcal{Y}$,

$$\max_{\boldsymbol{y} \in \mathcal{Y}} \boldsymbol{x}^{\top} \mathbf{A} \boldsymbol{y} - v \geq \kappa\|\boldsymbol{x} - \Pi_{\mathcal{X}^{\star}}(\boldsymbol{x})\|,$$

$$v - \min_{\boldsymbol{x} \in \mathcal{X}} \boldsymbol{x}^{\top} \mathbf{A} \boldsymbol{y} \geq \kappa\|\boldsymbol{y} - \Pi_{\mathcal{Y}^{\star}}(\boldsymbol{y})\|.$$

We focus on the first inequality, which is with respect to Player $x$. We let $\mathcal{X} := \{ \boldsymbol{x} \in \mathbb{R}^n : \boldsymbol{c}_i^{\top} \boldsymbol{x} \leq b_i \quad \forall i \in [\ell_x] \}$, where $\ell_x$ denotes the number of vertices of $\mathcal{X}$. We also let $\boldsymbol{o}_j := \mathbf{A}\boldsymbol{y}_j$, where $\boldsymbol{y}_j$ denotes the $j$th vertex of $\mathcal{Y}$; for simplicity, we will denote by $k_y \in \mathbb{N}$ the number of vertices of $\mathcal{Y}$. We consider a fixed $\boldsymbol{x} \in \mathcal{X} \setminus \mathcal{X}^{\star}$ and $\boldsymbol{x}^{\star} = \Pi_{\mathcal{X}^{\star}}(\boldsymbol{x})$.

---

[6]Wei et al. [2021] impose (40) only for points $\boldsymbol{z} \in \mathcal{Z} \setminus \mathcal{Z}^{\star}$, which is easily seen to be equivalent.

It is easy to see that the set of optimal strategies for Player $x$, $\mathcal{X}^\star \coloneqq \{x \in \mathcal{X} : \max_{y \in \mathcal{Y}}\langle x, \mathbf{A}y\rangle \leq v\}$, can be expressed as

$$\mathcal{X}^\star \coloneqq \left\{ x \in \mathbb{R}^n : c_i^\top x \leq b_i, o_j^\top x \leq v \quad \forall (i,j) \in [\ell_x] \times [k_y] \right\}.$$

Indeed, any point $y \in \mathcal{Y}$ is a convex combination of the vertices of $\mathcal{Y}$, and the converse direction is also obvious. A feasibility constraint $i \in [\ell_x]$ is said to be *tight* if $c_i^\top x^\star = b_i$; similarly, an optimality constraint $j \in [k_y]$ is tight if $o_j^\top x^\star = v$. We let $L_x = L_x(x^\star) \subseteq [\ell_x]$ be the set of tight feasibility constraints and $K_y = K_y(x^\star) \subseteq [k_y]$ be the set of tight optimality constraints. We can assume without any loss that $L_x, K_y \neq \emptyset$. It is well-known (*e.g.*, [Rockafellar, 2015]) that the *normal cone* of $\mathcal{X}^\star$ at $x^\star$ with respect to $\mathcal{X}^\star$ can be expressed as

$$N_{x^\star} \coloneqq \left\{ \sum_{i \in L_x} p_i c_i + \sum_{j \in K_y} q_j o_j \quad \forall (p, q) \in \mathbb{R}_{\geq 0}^{L_x} \times \mathbb{R}_{\geq 0}^{K_y} \right\}.$$

Wei et al. [2021] also define $M_{x^\star} \subseteq N_{x^\star}$ as

$$N_{x^\star} \cap \left\{ c_i^\top x \leq 0 \quad \forall i \in L_x \right\}.$$

Now, the main parameter of interest that relates to Definition 1.3 in the analysis of Wei et al. [2021] stems from the following quantity.

**Definition C.11.** We let $C \in \mathbb{R}_{>0}$ be defined as the infimum over $(0, \infty)$ so that

$$\left\{ \sum_{i \in L_x} p_i c_i + \sum_{j \in K_y} q_j o_j, 0 \leq p_i, q_j \leq C \right\} \supseteq M_{x^\star} \cap \mathcal{B}_\infty, \tag{43}$$

where $\mathcal{B}_\infty \subseteq \mathbb{R}^n$ is the set of points with $\ell_\infty$ norm upper bounded by 1.

By definition of $M_{x^\star}$, it is evident that there always exists a finite problem-dependent parameter $C \in \mathbb{R}_{>0}$ such that Definition C.11 is satisfied. It is then not hard to show that

$$\max_{y \in \mathcal{Y}} x^\top \mathbf{A} y - v \geq \frac{1}{C|K_y|}\|x - \Pi_\mathcal{X}(x^\star)\|.$$

Assuming that the number of vertices is polynomial in the dimensions,[7] this shows that Definition C.11 essentially captures the complexity of satisfying Definition 1.3. As we explained earlier in Section 3.1, the constraint matrix of the linear program induced by Definition C.11 depends both on the payoff matrix $\mathbf{A}$ as well as the set of constraints. It is thus unclear how to use existing results in the model of smoothed complexity [Dunagan et al., 2011] to bound $C$. The second and more important challenge revolves around the fact that Definition C.11 depends solely on the tight constraints of the optimal solution, which in turn depends on the randomness of $\mathbf{A}$. Under our characterization, the latter challenge was addressed earlier in Section 3.3.

Continuing for `OMWU`, we again rely on the analysis of Wei et al. [2021], which relates the rate of convergence of `OMWU` to three quantities. The first one [Wei et al., 2021, Definition 3] is similar to Definition C.9, but with the difference that the maximization is now constrained to be over points whose support is a subset of the support of the equilibrium; namely,

$$\kappa_x \coloneqq \min_{x \in \mathcal{X} \setminus \{x^\star\}} \max_{y \in \mathcal{V}^\star(\mathcal{Y})} \frac{\langle x - x^\star, \mathbf{A}y\rangle}{\|x - x^\star\|_1}, \tag{44}$$

where $\mathcal{V}^\star(\mathcal{Y}) \coloneqq \{y \in \Delta^m : \mathrm{supp}(y) \subseteq \mathrm{supp}(y^\star)\}$. A symmetric definition is to be considered with respect to Player $y$. To connect this to (8), we note that, when $y \in \mathcal{V}^\star(\mathcal{Y})$, $\langle x^\star, \mathbf{A}y\rangle = v$. We are thus left to lower bound $\max_y \langle x, \mathbf{A}y\rangle - v$ in terms of $\|x - x^\star\|_1$, but under the constraint that $y \in \mathcal{V}^\star(\mathcal{Y})$. An inspection of our proof of Theorem 3.6 (and in particular the proof of (8)) reveals that its conclusion holds even when the maximization is subject to the above constraint, and so our analysis

---

[7]In fact, by virtue of Carathéodory's theorem, one can refine Definition C.11 so that this holds even when the number of vertices is exponential in the dimensions. Namely, a point $v \in M_{x^\star} \cap \mathcal{B}_\infty$ can be written as the conical combination of at most $n$ of the vectors describing the cone in (43), thereby maintaining feasibility. This observation can be used to refine the (worst-case) analysis of Wei et al. [2021] to, for example, extensive-form games wherein the number of vertices is typically exponential in the dimensions.

immediately lower bounds (44) as well. The second quantity introduced by Wei et al. [2021, Definition 2] corresponds exactly to Item 2, which was bounded in Proposition 3.8. The third quantity [Wei et al., 2021, Definition 4] is where the exponential overhead is introduced. Namely, the iteration complexity of OMWU in their analysis depends on $\exp\left(\min(\alpha_P(\mathbf{A}), \alpha_D(\mathbf{A}))^{-1}\right)$, where we recall the definition in Item 1.[8] Unfortunately, *for any game*, it holds that $\alpha_P(\mathbf{A}) \leq 1/n$ and $\alpha_D(\mathbf{A}) \leq 1/m$, and so even if the geometry of the problem is favorable, the obtained bound is exponential. (The reason the above quantity is crucial in their analysis is because it lower bounds the probability of playing any action through the trajectory of OMWU.) Nevertheless, using Proposition 3.10, our analysis provides instead a bound of $\exp(\mathrm{poly}(n, m, 1/\sigma))$ with high probability, which is still a major improvement over the worst-case bound of Wei et al. [2021], which can be doubly exponential in the number of bits $L$ describing the game—one can easily make sure that $\alpha_P(\mathbf{A}) \approx 1/2^L$ (Proposition 3.1).

Next, for EGDA, Tseng [1995] established linear convergence under the error bound

$$\mathrm{dist}(\boldsymbol{z}, \boldsymbol{z}^\star) \leq \tau \|\boldsymbol{z} - \Pi_{\mathcal{Z}}(\boldsymbol{z} - \eta F(\boldsymbol{z}))\|$$

for some $\tau > 0$ and a suitable $\eta > 0$ [Tseng, 1995, Corollary 3.3]. It is easy to make the following connection.

**Lemma C.12.** *It holds that* $\Phi(\boldsymbol{z}) \leq \frac{2}{\eta}\|\boldsymbol{z} - \Pi_{\mathcal{Z}}(\boldsymbol{z} - \eta F(\boldsymbol{z}))\|$.

*Proof.* Indeed, by the first-order optimality condition for the optimization problem associated with

$$\boldsymbol{z}' := \Pi_{\mathcal{Z}}(\boldsymbol{z} - \eta F(\boldsymbol{z})) = \arg\min_{\boldsymbol{z}' \in \mathcal{Z}} \left\{ \|\boldsymbol{z}' - (\boldsymbol{z} - \eta F(\boldsymbol{z}))\|^2 := h(\boldsymbol{z}') \right\},$$

we get $\langle \widehat{\boldsymbol{z}} - \boldsymbol{z}', \nabla h(\boldsymbol{z}') \rangle \geq 0$ for any $\widehat{\boldsymbol{z}} \in \mathcal{Z}$, or equivalently, $\min_{\widehat{\boldsymbol{z}} \in \mathcal{Z}} \langle \widehat{\boldsymbol{z}} - \boldsymbol{z}', \boldsymbol{z}' - \boldsymbol{z} + \eta F(\boldsymbol{z}) \rangle \geq 0$. Observing that $\min_{\widehat{\boldsymbol{z}} \in \mathcal{Z}} \langle \widehat{\boldsymbol{z}}, F(\boldsymbol{z}) \rangle = -\Phi(\boldsymbol{z})$ and bounding

$$\langle \boldsymbol{z} - \boldsymbol{z}', \widehat{\boldsymbol{z}} - \boldsymbol{z}' \rangle \geq -\|\boldsymbol{z} - \boldsymbol{z}'\|\|\widehat{\boldsymbol{z}} - \boldsymbol{z}'\| \geq -D_{\mathcal{Z}}\|\boldsymbol{z} - \boldsymbol{z}'\| = -2\|\boldsymbol{z} - \Pi_{\mathcal{Z}}(\boldsymbol{z} - \eta F(\boldsymbol{z}))\|$$

leads to the claim. $\square$

It can thus be shown that Definition 1.3 is again sufficient to dictate the rate of convergence of EGDA.

Finally, for IterSmooth, Gilpin et al. [2012] introduced a "condition measure" of the payoff matrix $\mathbf{A}$, which in fact corresponds precisely to Definition 1.3. Thus, Theorem 1.2 with respect to IterSmooth follows readily from [Gilpin et al., 2012, Theorem 2].

### C.4  Proof of Theorem 4.2

Finally, we conclude with the proof of Theorem 4.2, which is restated below.

**Theorem 4.2.** *Any $\delta$-support-stable game (per Definition 4.1) satisfies the error bound for any sufficiently small modulus*

$$\kappa \geq \mathrm{poly}\left(\frac{1}{n}, \frac{1}{m}, \delta\right).$$

*Proof of Theorem 4.2.* We treat each parameter separately.

- Let us start from $\beta_P(\mathbf{A})$ (Item 2). We let $j' \in \arg\min_{j \in \overline{N}}(v - \langle \boldsymbol{x}_B^\star, \mathbf{A}_{B,j} \rangle)$, where we assume that $\overline{N} \neq \emptyset$. We consider a perturbed matrix $\mathbf{A}'$ such that

$$\mathbf{A}'_{i,j} = \begin{cases} \mathbf{A}_{i,j} - \beta_P(\mathbf{A}) & \text{if } i \in B, j = j', \\ \mathbf{A}_{i,j} & \text{otherwise.} \end{cases}$$

---

[8]More specifically, the proof of Wei et al. [2021, Theorem 3] upper bounds the Kullback-Leibler divergence $\mathrm{KL}(\boldsymbol{z}^{(t)}, \boldsymbol{z}^\star)$ by a quantity that is at least as large as $\left(1 + \frac{15\eta^2 C_2}{32}\right)^{-t}$, where $C_2 \leq \exp\left(\min(\alpha_P(\mathbf{A}), \alpha_D(\mathbf{A}))^{-1}\right)$. Thus, to guarantee $\mathrm{KL}(\boldsymbol{z}^{(t)}, \boldsymbol{z}^\star) \leq \epsilon$ using the analysis of Wei et al. [2021] one needs at least $\log(1/\epsilon)/\log\left(1 + \frac{15\eta^2 C_2}{32}\right)$ iterations. When $C_2 \ll 1$, this grows with $1/C_2 \geq \exp\left(\min(\alpha_P(\mathbf{A}), \alpha_D(\mathbf{A}))\right)$.

Then, the game described by $\mathbf{A}'$ cannot be non-degenerate with the same support as $\mathbf{A}$. Indeed, in the contrary case it would follow that the (unique) equilibrium $(\boldsymbol{x}_B^\star, \boldsymbol{y}_N^\star)$ remains the same since $\mathbf{A}'_{B,N} = \mathbf{A}_{B,N}$. But then, $v - \langle \boldsymbol{x}_B^\star, \mathbf{A}'_{B,j'} \rangle = v - \langle \boldsymbol{x}_B^\star, \mathbf{A}_{B,j'} \rangle - \beta_P(\mathbf{A}) = 0$, by definition of $j'$ and $\beta_P(\mathbf{A})$, which is a contradiction. Further, $\|\mathbf{A} - \mathbf{A}'\| = \beta_P(\mathbf{A})$. In turn, this implies that $\delta \leq \beta_P(\mathbf{A})$. Similar reasoning yields that $\delta \leq \beta_D(\mathbf{A})$.

- Continuing for $\gamma_P(\mathbf{A})$ (Item 3), we assume that $\widetilde{B}, \widetilde{N} \neq \emptyset$. We let $\mathbf{U}\boldsymbol{\Sigma}\mathbf{V}^\top$ be a singular value decomposition (SVD) of $\mathbf{Q}$. Then, a perturbation to $\mathbf{Q}$ of the form $\mathbf{U}\mathrm{diag}(0, 0, \ldots, \sigma_{\min}(\mathbf{Q}))\mathbf{V}^\top$ leads to a singular matrix $\mathbf{Q}'$, which cannot be the case if the perturbed game is non-degenerate with the same support. This perturbation can be cast in terms of $\mathbf{A}'_{B,N}$ through transformation $\mathbf{T}$ in (5). This lower bounds $\sigma_{\min}(\mathbf{Q})$ in terms of $\delta$, and Proposition C.4 can in turn lower bound $\gamma_P(\mathbf{A})$ in terms of $\sigma_{\min}(\mathbf{Q})$.

- Finally, we treat $\alpha_P(\mathbf{A})$ (Item 1). The non-trivial case is again when $\widetilde{B}, \widetilde{N} \neq \emptyset$. Let $i' \in \arg\min_{i \in B}(\boldsymbol{x}_i^\star)$. If $i' \in \widetilde{B}$, we define

$$\mathbb{R}^{\widetilde{B}} \ni \widetilde{\boldsymbol{x}}_i' = \begin{cases} 0 & \text{if } i = i', \\ \boldsymbol{x}_i^\star & \text{otherwise.} \end{cases}$$

We know that $\mathbf{Q}^\top \widetilde{\boldsymbol{x}}^\star = \boldsymbol{b}$. We then consider the perturbed vector $\boldsymbol{b}' := \mathbf{Q}^\top \widetilde{\boldsymbol{x}}'$. If the perturbed game was non-degenerate with the same support, it would follow that $(\widetilde{\boldsymbol{x}}', \cdot)$ is the unique equilibrium, which is a contradiction since $\widetilde{\boldsymbol{x}}_{i'} = 0$. Further, the norm of the perturbation $\|\boldsymbol{b} - \boldsymbol{b}'\|$ is upper bounded in terms of $\alpha_P(\mathbf{A})$, which can be again expressed in terms of $\mathbf{A}_{B,N}$ through transformation (5). Similarly, if $i' \notin \widetilde{B}$, we define

$$\mathbb{R}^{\widetilde{B}} \ni \widetilde{\boldsymbol{x}}_i' = \boldsymbol{x}_i^\star + \frac{\alpha_P(\mathbf{A})}{|\widetilde{B}|},$$

and we consider the perturbed vector $\boldsymbol{b}' := \mathbf{Q}^\top \widetilde{\boldsymbol{x}}'$. If the perturbed game was non-degenerate with the same support, it would follow that $(\widetilde{\boldsymbol{x}}', \cdot)$ is the unique equilibrium, which is a contradiction since $\sum_{i \in \widetilde{B}} \widetilde{\boldsymbol{x}}_i' = \sum_{i \in \widetilde{B}} \boldsymbol{x}_i^\star + \alpha_D(\mathbf{A}) = 1$. The norm of the perturbation is again upper bounded in terms of $\alpha_P(\mathbf{A})$. Overall, we have shown that $\delta \leq \alpha_P(\mathbf{A})\mathsf{poly}(n,m)$. Similar reasoning applies with respect to $\alpha_D(\mathbf{A})$. This completes the proof.

$\square$

