# OpenReview forum: "Convergence of $\text{log}(1/\epsilon)$ for Gradient-Based Algorithms in Zero-Sum Games without the Condition Number: A Smoothed Analysis"
_NeurIPS.cc/2024/Conference — NeurIPS 2024 poster_

### Official Review · Reviewer_RDh1 · 2024-07-10

**Soundness:** 3
**Presentation:** 3
**Contribution:** 3
**Rating:** 6
**Confidence:** 1

**Summary:**

The paper provides smoothed analysis results four gradient-based algorithms for two-player zeros-sum games: OGDA, OMWU, EGDA and IterSmooth. The considered setting assumes that the payoff matrix is injected by noise where each element of the noise matrix is i.i.d. Gaussian $\mathcal{N}(0, \sigma^2)$. In this setting, the linear convergence rate $O(\log(1/\epsilon))$ of three algorithms OGDA, EGDA and IterSmooth are proved to hold with probability $1- \frac{1}{mn}$, where $m$ and $n$ are dimensions of the action space of the two players. The proofs rely on a key insight that with high probability over the randomness of the noise, the payoff matrix (and hence the game) satisfies a particular error bound that gives rise to the $\log(1/\epsilon)$ convergence rate.

**Strengths:**

The paper offers novel techniques to analyze the reliability of existing gradient-based algorithms. These new techniques sidestep the difficulty on condition numbers-like quantity in previous analyses, and instead focus on the geometric characteristics of (the equilibrium of) the game. This seems highly interesting and significant.

**Weaknesses:**

I find the writing hard to understand. Theorems, lemmas and technical results are often discussed before they are explained and/or formally defined.  In particular, the definition of the matrix $Q$ in Equation (5) is quite confusing as it is not worded as a definition at all. The vector $b, c$ and value $d$ also seem to come out of no where.

**Questions:**

1. In the presentation of section 3.2, the matrix $Q$ (and subsequently $\tilde{x}$ and $\tilde{y}$)  is defined based on a particular pair of indexes $(i, j) \in B \times N$. Are all subsequent claims hold for all $(i, j)$?

2. Is there a reason for the missing proof of claim C.2?

**Limitations:**

The work is of mathematical nature and has no societal impact.

---

> ### Author Rebuttal · Authors · 2024-08-05
>
> We thank the reviewer for their time and service.
>
> *In the presentation of section 3.2, the matrix $Q$ is defined based on a particular pair of indexes $(i,j) \in B \times N$. Are all subsequent claims hold for all $(i,j)$?*
>
> Matrix $Q$ is indeed defined with respect to a certain pair $(i,j) \in B \times N$, and all subsequent claims hold for any such $(i, j)$.
>
> *Is there a reason for the missing proof of claim C.2?*
>
> The proof of Claim C.2 is in the paragraph just before Claim C.2. We are happy to transfer the proof after the claim if the reviewer believes the current version can cause confusion.
>
> *The definition of the matrix $Q$  in Equation (5) is quite confusing as it is not worded as a definition at all. The vector $b, c$ and value $d$ also seem to come out of no where.*
>
> Those quantities are defined in Eq. (10) of the appendix. Their exact definition is not important for the purpose of the main body, but we will make sure to include Eq. (10) in the main body of the revision as we see how the current version can cause some confusion.
>
> Let us know if we can further make the writing easier to understand.

---

> > ### Comment · Reviewer_RDh1 · 2024-08-08
> >
> > Thanks for the clarification. I will keep my score.

---

### Official Review · Reviewer_NLN1 · 2024-07-12

**Soundness:** 3
**Presentation:** 3
**Contribution:** 3
**Rating:** 7
**Confidence:** 1

**Summary:**

The paper is concerned with studying the convergence of some state-of-the-art gradient-based algorithms for solving zero-sum games. For these algorithms, it is known that in the worst case, the number of their iterations grows polynomially in 1/e, where e is the error bound.

The paper shows that for many of the aforementioned algorithms, their smoothed complexity is polynomial in the sense that their number of iterations grows polynomially in log(1/e).

**Strengths:**

1. The paper studies state-of-art algorithms that have significant important in ML applications.
2. The paper provides meaningful justifications as to why some of the gradient-based algorithms for zero-sum games perform well in practice.

**Weaknesses:**

No noted weaknesses. This seems to be a very solid paper that's very relevant to the scope of NeurIPS.

**Questions:**

No questions.

**Limitations:**

No limitations found.

---

> ### Author Rebuttal · Authors · 2024-08-05
>
> We thank the reviewer for their time and service.

---

### Official Review · Reviewer_La2m · 2024-07-12

**Soundness:** 3
**Presentation:** 3
**Contribution:** 2
**Rating:** 6
**Confidence:** 3

**Summary:**

This paper performs a smoothed analysis for zero-sum games. Existing convergence rate guaratees of gradient-based algorithms often depend on condition number-like quantities which can be exponential in dimension. This paper shows that for the average case or smoothed case (as opposed to worst-case) the error condition constant can be polynomial in dimension. This validates practically observed success of gradient-based algorithms to solve zero-sum games.

The paper also discusses the relation between $\delta$-stable games and error-bound condition constant.

**Strengths:**

The main strength of the paper is Theorem 1.4 which shows that, in the smoothed case, the error-bound coefficient is polynomial in problem dimensions. This deals with an important problem and bridges the gap between theoretical rates and success of gradient-based algorithms in real life to solve zero-sum games.

The paper is easy to read.

**Weaknesses:**

1. The paper lacks adequate comparison with existing literature. There has been considerable research on smoothed analysis of optimization problems (see [1] for example). Since standard (worst-case) analysis of gradient-based algorithms like OGDA, EGDA, etc. to solve (1) is quite similar to analysis of gradient-based algorithms to solve constrained convex optimization problem, I expect the novelty required to deal with the smoothed version of (1) should be pretty similar to the established techniques to deal with smoothed analysis of constrained convex optimization problem. If this is not true, I request you to provide a detailed discussion delineating the two scenarios and highlighting the novelties required in this paper on top of smoothed analysis of convex minimization problems.

2. Convergence in terms of euclidean distance instead of duality gap should be less emphasized as contribution/novelty as these type of results are already known for gradient-based algorithms.

Minor Suggestions:

3. The paper abstract and possibly title should be reflective of the fact that Theorem 1.4 is the main message of the paper (instead of convergence rates) as all the other results of the paper readily follows from Theorem 1.4.

4. An example where the the constant $\kappa$ depends exponentially on $m,n$ could be helpful.

Reason for my score: Theorem 1.4 seems to be the main contribution - the proof of which mostly requires change of variables and well-known concentration results for Gaussian random variables. This result by itself, although a very nice result, seems a little inadequate for a NeurIPS level paper. But I am on the fence here and I may change my score depending on answer to the first question under Weakness section.

[1]Cunha, Leonardo, Gauthier Gidel, Fabian Pedregosa, Damien Scieur, and Courtney Paquette. "Only tails matter: Average-case universality and robustness in the convex regime." In International Conference on Machine Learning, pp. 4474-4491. PMLR, 2022.

**Questions:**

See Weakness section.

---

> ### Author Rebuttal · Authors · 2024-08-05
>
> We thank the reviewer for their time and service.
>
> *The paper lacks adequate comparison with existing literature. There has been considerable research on smoothed analysis of optimization problems (see [1] for example).*
>
> We will make sure to cite and discuss [1] (and some of the references therein) in the revised version. There are many crucial differences between our results and [1].
>
> - First, [1] focuses on certain convex quadratic problems while we examine zero-sum games.
> - The perturbation model between our paper and [1] are also different, as the latter has an average-case flavor (parameterized by the concentration of eigenvalues of a certain matrix) while we work in the usual smoothed complexity model of Spielman and Teng.
> - Our main result establishes an iteration complexity scaling with $\log(1/\epsilon)$, while [1] obtains a complexity polynomial in $1/\epsilon$ (unless there is strong convexity, which is not the case in our problem).
> - Moreover, the algorithms considered in [1] are distinct from ours, although they are also gradient-based, and the techniques are also very different. In particular, [1] assumes (see Problem 2.1 in that paper) that the underlying randomization is independent of the optimal solution. On the other hand, as we stress throughout our paper, the fact that in our setting the equilibrium depends on the randomization is the main technical challenge.
>
> Overall, the technical challenges we faced are very different from the ones in [1], and we believe that the papers are generally orthogonal. We will include the above discussion in the revision.
>
>
> *Convergence in terms of euclidean distance instead of duality gap should be less emphasized as contribution/novelty as these type of results are already known for gradient-based algorithms*
>
> The contribution/novelty of our results in terms of Euclidean distance is that they provide the first polynomial bounds in the smoothed complexity model. Many results in terms of Euclidean distance were known, as the reviewer points out and as we highlight throughout the paper, but they had to rely on exponentially large constants. We believe that this improvement–which is obtained in conjunction with those known results–is worth highlighting as an important contribution.
>
>
> *​​The paper abstract and possibly title should be reflective of the fact that Theorem 1.4 is the main message of the paper (instead of convergence rates) as all the other results of the paper readily follows from Theorem 1.4.*
>
> We will expand the abstract to reflect the fact that Theorem 1.4 is the main technical contribution.
>
> *An example where the constant $\kappa$ depends exponentially on $m, n$ could be helpful.*
>
> Starting from the $3 \times 3$ example of Proposition 3.1, one can make $\kappa$ to be inversely exponential in $m, n$ by suitable selecting $\gamma = \gamma(n, m)$ (and filling the rest of the matrix according to the identity matrix).

---

> > ### Author Response · Authors · 2024-08-12
> >
> > We thank the reviewer again for the valuable feedback. Given that the discussion period comes to an end tomorrow, we wanted to make sure that our response above adequately addressed the reviewer's concern regarding comparison with prior work.

---

> > > ### Comment · Reviewer_La2m · 2024-08-12
> > >
> > > Thanks for the clarification. I have increased my score.

---

### Official Review · Reviewer_uKXc · 2024-07-13

**Soundness:** 3
**Presentation:** 3
**Contribution:** 3
**Rating:** 6
**Confidence:** 3

**Summary:**

This paper studies smoothed analysis of gradient-based algorithms for computing equilibria in zero-sum games. In general, regret minimization can be used to compute an $\epsilon$-equilibrium in a number of iterations polynomial in $1/\epsilon$. If inverse polynomial or better precision in $\epsilon$ is desired, this iteration count becomes prohibitively large. In this case LP solvers based on interior point methods can be used, which have iteration counts of $\log(1/\epsilon)$, but high per-iteration complexity and memory requirements. This paper shows that several standard gradient-based algorithms converge to an $\epsilon$-approximate equilibrium in a number of iterations polynomial in $\log(1/\epsilon)$ in the setting of smoothed analysis. In particular the runtime also depends polynomially on $1/\sigma$, where $\sigma$ is the variance of the smoothing noise.

The general approach taken in this paper is to prove that, for a smoothed zero-sum game, the duality gap of a pair of strategies is with high probability at most polynomially (in the size of the game  $nm$ and the noise variance $\sigma$) smaller than the $\ell_2$ distance to the equilibrium strategy pair. Existing analysis of several standard gradient descent methods then imply convergence to an $\epsilon$-equilibrium in time polynomial in $nm,1/\sigma,\log(1/\epsilon)$.

**Strengths:**

1. This paper provides, to my knowledge, the first theoretical justification for using gradient-based methods as equilibrium solvers in zero sum games in the high-precision regime, i.e. where $\epsilon$ is inverse polynomial in the size of the game.
2. The approach via smoothed analysis is quite generic: it revolves around structural properties of the game relating distance to the equilibrium and the duality gap. This structural property appears in the analysis of several gradient-based algorithms, and so the results can be applied to directly obtain $\log(1/\epsilon)$ convergence to $\epsilon$-equilibria.

**Weaknesses:**

The paper relies heavily on techniques for smoothed analysis of linear programming developed by Spielman and Teng. This is in some sense to be expected due to the strong connections between zero-sum games and linear-programming. As the authors point out, the smoothed analysis of gradient-based methods is well-known in the unconstrained min-max setting, and the main problem is that for zero-sum games the strategies must be constrained to be probability distributions.


Clarity:

Line 260: The introduction of $Q$, $d$, and $c$ here as implicitly defined by equation (5) was not very illuminating. Going to the appendix to see the full definition eventually resolves this, but this part could be better written.

Line 774 and 776:  Refs to Definition 3.1 should probably be refs to Theorem 3.6.

**Questions:**

Could you clarify precisely how gradient based methods in the smoothed setting improve over, say interior point methods which also achieve $\log(1/\epsilon)$ iteration complexity? Probably you can claim lower space requirements, but is there anything else to say?

**Limitations:**

Yes.

---

> ### Author Rebuttal · Authors · 2024-08-05
>
> We thank the reviewer for their time and service.
>
> We also thank the reviewer for pointing out the issues with clarity. We will make sure to incorporate the suggestions in the revision.
>
> *Could you clarify precisely how gradient based methods in the smoothed setting improve over, say interior point methods which also achieve $\log(1/\epsilon)$ iteration complexity? Probably you can claim lower space requirements, but is there anything else to say?*
>
> The two main aspects on which gradient-based algorithms, such as OGD, are more appealing than interior-point methods are the per-iteration complexity and the memory requirements. An algorithm such as OGD requires a single matrix-vector product per iteration; this is nearly linear for sparse matrices, and can be even smaller when the game-matrix is more structured (for example, low-rank). The memory requirements of OGD are also minimal, as the reviewer points out. On the other hand, interior-point methods require solving a linear system in each iteration, which can be prohibitive in large games. In those regards gradient-based algorithms are more compelling.

---

> > ### Comment · Reviewer_uKXc · 2024-08-13
> >
> > Thanks for the response! I will maintain my score and continue to recommend acceptance.

---

### Official Review · Reviewer_L1q2 · 2024-07-14

**Soundness:** 3
**Presentation:** 3
**Contribution:** 3
**Rating:** 6
**Confidence:** 4

**Summary:**

The authors apply the smoothed analysis framework predominantly studied by (Spielman and Teng'04) to some common sequential algorithms for learning the Nash equilibria in zero-sum bimatrix games. To this end, they look at EGDA, OGDA, OMWU and IterSmooth. They look at these algorithms that have known last-iterate convergence properties, albeit with game dependent constants or do not have linear rates in the worst case. Their main techniques are to show important constants and condition numbers have polynomial dependence on the game dimensions and the noise parameter.

**Strengths:**

This is one way to understand the average case performance of some common algorithms for solving bimatrix games. Their results instill some sense of confidence in these algorithms as their average case performance is polynomial in the dimensions and has a linear last-iterate behavior, except OMWU, which perhaps indicates some intrinsic intractability with respect to the best obtainable constants still being game dependent!

**Weaknesses:**

1) I think the authors could perhaps provide more justification for using smoothed analysis in this context. In the standard sense, it is about understanding perturbation of worst case instances (for simplex). For games, often we do not clearly know the worst case game matrices for certain algorithms.

2) Some assumptions related to perturbations of the A matrix, that is in definition 4.1 could be restrictive, in that the support of the equilibrium doesn't change. Can the authors comment on whether this can be relaxed to a certain extent?

**Questions:**

Can the authors comment if the smoothed analysis could give insights going beyond bimatrix games, such as the performance of the algorithms studied in this paper but for convex-concave settings or for low-rank bimatrix non-zero sum games?

Minor:
Typo in page 8, line 395 ...of the error bond->...bound.

**Limitations:**

yes

---

> ### Author Rebuttal · Authors · 2024-08-05
>
> We thank the reviewer for their time and service.
>
> *Can the authors comment if the smoothed analysis could give insights going beyond bimatrix games, such as the performance of the algorithms studied in this paper but for convex-concave settings or for low-rank bimatrix non-zero sum games?*
>
> This is an interesting question. Beyond (two-player) matrix games, perhaps the most natural next step would concern polymatrix zero-sum games, for which we suspect that similar results should apply. For general convex-concave games, it is not clear to us how one should even define the perturbed instance, unless the underlying function is more structured. When it comes to nonzero-sum bimatrix games, even under a low-rank constraint computing Nash equilibria is hard (unless the rank is 1), and so iterative algorithms instead converge in a time-average sense to a so-called coarse correlated equilibrium (CCE). It is unclear whether the framework of smoothed complexity can lead to meaningful improvements when it comes to convergence to CCE. On the other hand, it seems plausible that our results can be extended to rank-1 games.
>
> *Some assumptions related to perturbations of the A matrix, that is in definition 4.1 could be restrictive, in that the support of the equilibrium doesn't change. Can the authors comment on whether this can be relaxed to a certain extent?*
>
> There are other natural notions of perturbation stability one could consider in the context of Section 4; for example, that the equilibrium of the perturbed game must be $\delta$-close to the equilibrium of the original game. It is not clear whether a result such as Theorem 4.2 applies to that notion, but we believe this is an interesting question for future work. One technical challenge we should point out here is that our characterization of Theorem 3.6 does not handle multiplicity of equilibria, which appears to be relevant for other notions of perturbation stability.
>
> *I think the authors could perhaps provide more justification for using smoothed analysis in this context.*
>
> Besides the usual justification for performing smoothed analysis in the context of linear programming and optimization, which applies to zero-sum games as well, we believe that in game-theoretic problems smoothed complexity is even more relevant since there is often noise/imprecision when specifying the players’ utilities. This puts into question some worst-case pathological examples, such as the one specified in Proposition 3.1. Those are the type of examples for which algorithms such as OGD perform poorly in practice.
>
> Finally, we thank the reviewer for spotting the typo.

---

> > ### Comment · Reviewer_L1q2 · 2024-08-09
> > **Response to Rebuttal**
> >
> > I thank the authors for their rebuttal and I will maintain my existing positive score.

---

### Decision · Program_Chairs · 2024-09-25

**Decision:**

Accept (poster)

**Comment:**

This paper studies the smoothed analysis of gradient-based algorithms for computing equilibria in bilinear two-player zero-sum games over simplexes in a high-precision regime. The existing algorithms covered in this paper have known last-iterate worst-case convergence rate but might suffer from an undesirable dependency on the dimension of the matrix, as discussed via a toy example from Proposition 3.1 in this paper.

Motivated by this issue, the paper studies the smoothed analysis of some gradient-based algorithms for computing equilibria in zero-sum games and shows that their complexity depends on a polynomial in the dimension, $\log(1/\epsilon)$, and $1/\sigma$, where $\sigma$ measures the magnitude of the smoothing perturbation.

This paper received unanimous support, has gone through the discussions, and is hence recommended for acceptance.

In the camera-ready version, it would be more illuminating if some proof-of-concept experimental results (e.g., using the toy example from Proposition 3.1) could be included to support the theoretical results.